# PLAN FOR SPEED: DILATED SCHEDULING FOR MASKED DIFFUSION LANGUAGE MODELS

## ABSTRACT

Masked diffusion language models (MDLMs) promise fast, non-autoregressive text generation, yet existing samplers, which pick tokens to unmask based on model confidence, ignore interactions when unmasking multiple positions in parallel and effectively reduce to slow, autoregressive behavior. We propose the Dilated Unmasking Scheduler (DUS), an inference-only, planner-model-free method that partitions sequence positions into non-adjacent dilated groups and unmasked them in parallel so as to minimize an upper bound on joint entropy gain at each denoising step. By explicitly trading off the number of network calls against generation quality, DUS recovers most of the performance lost under traditional parallel unmasking strategies. Across math (GSM8K, MATH500), code (HumanEval, MBPP), general-knowledge (BBH, MMLU-Pro), and instruction following (IFEval) benchmarks, DUS outperforms confidence-based planners, without modifying the underlying denoiser, and reveals the true speed-quality frontier of MDLMs.

## 1 INTRODUCTION

Diffusion-based language models have emerged as a promising alternative to traditional autoregressive (AR) large language models (Grattafiori et al., 2024; Yang et al., 2025), offering potential advantages in parallel generation and controllability (Campbell et al., 2022; Lou et al., 2024; Sahoo et al., 2024). In the discrete diffusion paradigm for text, generation proceeds as an iterative denoising process: starting from a fully noised (e.g., masked) sequence, the model gradually reconstructs the original text over multiple timesteps. While this enables any-order generation, better quality requires one denoising pass per token, which leads to slow inference, as generating a sequence of length $G$ typically entails $\mathcal{O}(G)$ denoiser calls.

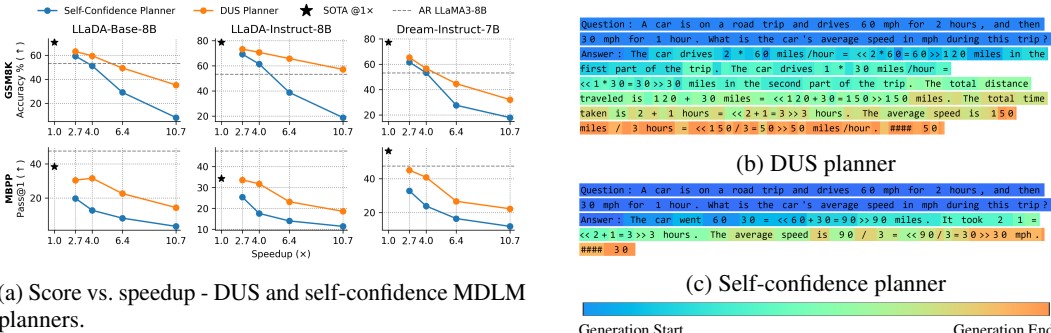

(a) Score vs. speedup - DUS and self-confidence MDLM planners.

(b) DUS planner

(c) Self-confidence planner

Figure 1: (a) Trade-off between inference speedup and task accuracy on math (GSM8K, MATH500) and code (HumanEval, MBPP) benchmarks. Solid orange curves show DUS planner results; solid blue curves show self-confidence planner results. The star ($\star$) marks the single-token, token-by-token SOTA baseline, and dashed gray lines denote AR LLaMA3-8B performance under the same protocol (Nie et al., 2025b; Ye et al., 2025b; Grattafiori et al., 2024). (b), (c) Chain-of-thought on a GSM8K example with block size $B = 32$ (speedup $6.4\times$). Token shading (blue→orange) encodes the unmasking iteration. (b), DUS, generates a full, coherent reasoning trace; (c), self-confidence, truncates its chain-of-thought prematurely.

AR LLMs have traditionally generated text token-by-token, leading to linear inference latency. Speculative decoding tackles this by first using a lightweight "draft" model to propose multiple tokens in parallel, then having the full AR model verify them - achieving speedups without retraining or architecture changes (Leviathan et al., 2023; Xia et al., 2022). By contrast, diffusion-based LMs reconstruct the entire sequence simultaneously: although naive masked-denoising still incurs $\mathcal{O}(G)$ passes, it inherently supports any-order and fully parallel generation, laying the groundwork for inference schedules that reduce denoiser calls below linear time.

Existing strategies to accelerate diffusion sampling either introduce heuristic planners that select tokens to unmask based on confidence or entropy, or employ semi-AR blockwise generation, which divides the sequence into contiguous spans and applies parallel diffusion within each block to preserve global coherence (Ye et al., 2025b; Arriola et al., 2025; Nie et al., 2025b). However, selecting a planner can be shortsighted and prone to error propagation, and semi-AR diffusion still incurs $\mathcal{O}(B)$ denoiser calls per block of size $B$. Recent work has demonstrated the effectiveness of framing unmasking as an inference-time planning paradigm (Peng et al., 2025), while others have incorporated planning mechanisms directly into the ELBO training objective (Liu et al., 2025). Additional research addresses the computational inefficiency of standard token-by-token denoising by proposing non-uniform schedules (Park et al., 2024) or denoiser's signal based unmasking schemes (Ben-Hamu et al., 2025; Wu et al., 2025) enabling dynamic multiple tokens to be revealed in parallel.

In this work, we introduce the *Dilated Unmasking Scheduler* (DUS), a purely inference-time, model-agnostic planner that partitions each block of length $B$ into logarithmically many iterations by revealing tokens in a fixed dilation pattern. Under Markov assumption, this schedule minimizes the joint conditional entropy at each step, although reducing the number of denoiser calls from $\mathcal{O}(B)$ to $\mathcal{O}(\log B)$ without any retraining or extra planner modules. Our main contributions are:

- **Inference-Only, Model-Agnostic Decoding:** A drop-in strategy requiring zero modifications to model architecture or training.

- **Logarithmic Unmasking Schedule:** Deterministic, coarse-to-fine dilation that respects local context and minimizes joint entropy, in $\mathcal{O}(\log B)$ denoiser iterations.

- **Theoretical Guarantees:** We show that DUS approaches the joint entropy bound at each iteration under fast-mixing assumptions, compared to baselines with the same step budget.

- **Empirical Validation:** Extensive experiments on math (GSM8K (Cobbe et al., 2021), MATH500 (Lightman et al., 2023)), code (HumanEval (Chen et al., 2021), MBPP (Austin et al., 2021)), general-knowledge (BBH (Suzgun et al., 2022), MMLU-Pro (Wang et al., 2024)), and instruction following (IFEval (Zhou et al., 2023)) benchmarks with LLaDA-8B, Dream-7B, DiffuCoder-7B demonstrate up to an order-of-magnitude reduction in denoiser calls and consistent quality improvements over confidence-based planners.

The remainder of this paper is organized as follows. Section 2 presents the fundamentals of masked diffusion language models. In Section 3, masked diffusion framework is formalized, DUS is introduced, and theoretical analysis under a fast-mixing variable-length Markov chain (VLMC) assumption is given. In Section 4, we present experimental results on mathematical reasoning, code generation, and general knowledge chain-of-thought (COT) benchmarks. Finally, Section 5 concludes and outlines future directions.

## 2 RELATED WORK

Discrete diffusion sampling relies on a planner to decide which masked tokens to reveal at each reverse step (Liu et al., 2025). Early "self-planners" use the diffusion denoiser itself to rank positions by simple criteria: top-$k$ highest probability (confidence), lowest conditional entropy, or top-$k$ probability margin (the gap between the highest and second-highest scores) (Campbell et al., 2022; Sahoo et al., 2024; Kim et al., 2025).

Path Planning (P2) extends beyond these self-planners by incorporating an external guiding model - a pretrained BERT - to evaluate candidate token sets according to their output probabilities (Peng et al., 2025). P2 compares denoiser-guided selection, random sampling, and BERT-scored planners, finding that the learned BERT planner yields better scores in various experiments at the cost of auxiliary model calls.

Semi-AR block diffusion segments a long sequence into contiguous spans of length $B$, runs denoiser iterations in parallel within each block, and reveals blocks sequentially (Arriola et al., 2025; Nie et al., 2025b). This approach does not reduce the total number of denoiser passes from $\mathcal{O}(G)$ to $\mathcal{O}(G/B)$ (where $G$ is the generation length), it still incurs $\mathcal{O}(B)$ calls per block, and it typically requires a policy to decide block order, most often determined by data characteristics. While the first (Arriola et al., 2025) discuss both training and inference with semi-AR block diffusion, here we focus exclusively on inference.

At the 7-8 B-parameter scale, two masked diffusion language models-Dream-7B (Ye et al., 2025b) and LLaDA-8B (Nie et al., 2025b)-demonstrate the practical viability of fully parallel decoding. LLaDA-8B, trained on 2.3 T tokens, matches LLaMA3-8B (Grattafiori et al., 2024) across a broad suite of zero and few-shot evaluations, particularly on mathematical reasoning and Chinese, language benchmarks-while retaining any-order generation via its principled diffusion objective and efficient remasking schedules. Dream-7B, pretrained on 580B tokens with AR model initialization and context-adaptive token-level noise rescheduling, achieves strong performance on mathematics, coding, and planning tasks. Both models offer supervised fine-tuned (SFT) variants for instruction following and applying semi-AR decoding over predefined blocks.

DiffuCoder (Gong et al., 2025) is a 7B-parameter MDLM trained solely on 130 billion tokens of code, an effort that its authors claim frees the model from the strong left-to-right bias typical of text-trained diffusion LLMs. By adjusting sampling temperature, DiffuCoder dynamically controls its "causalness", allowing fully parallel decoding without resorting to semi-AR blocks. Furthermore, the introduction of a coupled-GRPO reinforcement learning step after pretraining further reduces residual AR tendencies and yields an improvement on code benchmarks without any semi-AR inference fallbacks.

Inference efficiency can also be improved orthogonally via Key-Value (KV) caching, where stable key and value tensors from transformer layers are reused across denoising steps. Recent works such as *FreeCache* (Hu et al., 2025), *Fast-dLLM* (Wu et al., 2025) and *dKV-Cache* (Ma et al., 2025) report up to $3\times$ speedups with minimal impact on generation quality. These approaches require additional memory to store cached activations, and in the case of FreeCache an auxiliary AR model is needed to guide diffusion.

Adaptive scheduling methods determine which tokens to unmask and when to do so during sampling, with the goal of improving inference speed while preserving generation quality. On the timestep axis, *Jump Your Steps* (JYS) selects a non-uniform subset of noise levels by estimating KL divergences between adjacent diffusion kernels - trading a small ELBO loss for significantly fewer denoiser calls (Park et al., 2024). Common discrete masked diffusion models, however, are trained with cross-entropy objectives rather than continuous score-based losses (Sahoo et al., 2024), making direct application of log-likelihood schedules less straightforward. At the token granularity, entropy-bounded (EB) sampler uses threshold to pick the largest set of tokens whose cumulative conditional entropy remains below a user-specified bound - achieving $2-3\times$ empirical speedups on coding and math benchmarks without retraining (Ben-Hamu et al., 2025) and a similar method introduce a confidence bounded (CB) adaptive unmasking showing matching results (Wu et al., 2025). This per-token heuristic offers practical gains but lacks formal iteration-complexity guarantees and does not fully account for token dependencies when unmasking in parallel.

At scale, both open-source and commercial systems illustrate the practical adoption of diffusion LLMs. Dream-7B and LLaDA-8B employ planner heuristics and blockwise (semi-AR) diffusion to rival AR baselines on text and code benchmarks (Ye et al., 2025a;b; Nie et al., 2025b), and recent offerings - DeepMind's Gemini Diffusion (Google DeepMind, 2025) and Inception Labs' Mercury (Labs, 2025) - further underscore the growing throughput and practical adoption of diffusion-based language models.

## 3 METHOD

This section introduces the foundational concepts of MDLMs, reviews existing unmasking planners, presents our DUS, and provides theoretical guarantees of its optimality.

## 3.1 NOTATION

Let $\mathcal{X} = \{X_1, \ldots, X_G\}$ be a sequence of random variables forming a stationary, ergodic, VLMC with fast-mixing properties. At each decoding iteration step $t$, a masked version of the sequence is denoted by $\mathcal{M} = \{M_1, \ldots, M_G\}$, where $M_i$ is replaced by $X_i$ if token $i$ has been unmasked, and $M_i = \texttt{[MASK]}$ otherwise. Let $\mathcal{S}_t \subset \{\mathcal{X}, \mathcal{M}\}$ denote the state at denoiser iteration $t$, i.e., the collection of all currently known (unmasked) tokens and remaining masked positions. Those are non-strict assumptions used for the analysis which aims to capture that, in many text domains, local dependencies dominate and correlations typically decay with distance, while known edge cases with stronger long-range structure are discussed at the end of this section.

Define a *planner* $\mathcal{P}_t$ that chooses $k \leq G$ candidate masked indices $\{i_1, \ldots, i_k\} \subset \{1, \ldots, G\}$ at iteration $t$, and a *denoiser* $\mathcal{D}_\theta$ that maps each $M_{i_j}$ to a prediction of $X_{i_j}$ conditioned on the current state

$$\hat{X}_{i_j} = \mathcal{D}_\theta(M_{i_j} \mid \mathcal{S}_t). \tag{1}$$

A natural goal in conditional generation is to produce sequences with high likelihood under the data distribution. MDLMs are trained by maximizing a masked cross-entropy ELBO that upper-bounds $(-\log p_\theta(x))$ (Sahoo et al., 2024; Shi et al., 2025; Campbell et al., 2022; Nie et al., 2025b). At inference time the parameters $\theta$ are fixed and the only remaining degree of freedom is the planner $\mathcal{P}$. For a given planner, running the sampler induces a distribution $q_{\theta,\mathcal{P}}(\mathcal{X} \mid S_0)$ over generated sequences given an initial state $S_0$ (e.g., a prompt), and compare planners via the joint conditional entropy

$$H_{\theta,\mathcal{P}}(\mathcal{X} \mid S_0) = \mathbb{E}_{q_{\theta,\mathcal{P}}}\big[-\log q_{\theta,\mathcal{P}}(\mathcal{X} \mid S_0)\big], \tag{2}$$

taking lower $H_{\theta,\mathcal{P}}(\mathcal{X} \mid S_0)$ as a proxy for higher-likelihood generations under fixed denoiser $\mathcal{D}_\theta$.

## 3.2 MDLM AS DENOISER AND PLANNER

Masked diffusion samplers can be decomposed into two core components:

1. **Denoiser.** A pretrained network $\mathcal{D}_\theta$ that, at each reverse diffusion step, takes a partially masked sequence and predicts the values of masked tokens given the current masked and unmasked context (Campbell et al., 2022; Austin et al., 2023).

2. **Planner.** A strategy $\mathcal{P}_t$ that, at each iteration $t$, selects which positions in the sequence to unmask next. Examples include ancestral sampling with random selection, confidence-based and entropy-based selection, and learned planners (Peng et al., 2025; Liu et al., 2025).

In large-scale discrete diffusion language models, inference is often implemented in a semi-AR, block-wise fashion - the sequence is partitioned into consecutive blocks of size $B$, and each block is iteratively unmasked and denoised before moving to the next (Nie et al., 2025b). Block diffusion variants train the denoiser by masking contiguous spans of length $B$ and recovering them jointly, which has been observed to improve local coherence, while other methods train on unmasking tokens across the whole sequence (Arriola et al., 2025).

The DUS method acts as an *inference-only* planner - the denoiser $\mathcal{D}_\theta$ is kept fixed, and only the sequence of unmasking decisions $\{\mathcal{P}_t\}$ is changed. As a result, DUS can be used as a drop-in planning strategy for standard masked diffusion models, including MDLM (Sahoo et al., 2024), MD4 (Shi et al., 2025), LLaDA (Nie et al., 2025b), Dream (Ye et al., 2025b;a), and DiffuCoder (Gong et al., 2025). For block-wise samplers with block size $B$, DUS reduces the number of denoiser evaluations per block from $\mathcal{O}(B)$ to $\mathcal{O}(\log B)$, and is orthogonal to engineering accelerations such as KV caching and parallel decoding.

## 3.3 DETAILED FORMULATION

Because the denoiser is trained to reconstruct masked tokens at each reverse step by minimizing a masked ELBO, it is not optimized to reveal all tokens in a single pass. Consequently, inference unfolds over multiple unmasking-denoising iterations (Campbell et al., 2022; Nie et al., 2025a).Large-scale diffusion LLMs typically partition the sequence into semi-autoregressive blocks of length $B$ and apply several denoiser iterations within each block before moving to the next (Nie et al., 2025b).

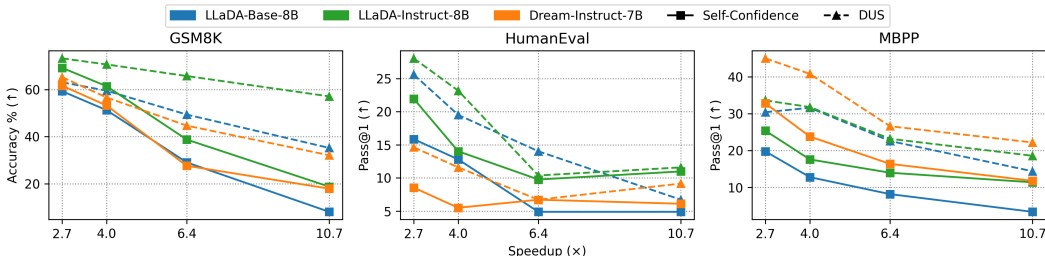

Figure 2: Experiments on GSM8K, HumanEval, and MBPP for variate speedup factors - defined by semi-AR inference block size. Higher score (Accuracy / Pass@1) is better. Each color represent a different model, while marker indicates the two planners tested - self-confidence ■; DUS ▲. Across all datasets and speedups, DUS achieves higher scores compared to the traditional planner.

At iteration $t$, let $\mathcal{S}_t$ denote the full state of the sequence: (1) all previously unmasked blocks, (2) the current block with a mixture of masked and unmasked tokens, and (3) the remaining future blocks that are still fully masked. Denote the indices of the current block by $\{b, \ldots, b+B-1\}$. The planner $\mathcal{P}_t$ selects a subset of indices $I_t = \{i_1, \ldots, i_k\} \subseteq \{b, \ldots, b+B-1\}$ to unmask. For a given state $\mathcal{S}_t$ and candidate index set $I_t$, a natural local objective is the block-level joint conditional entropy $H_\theta(X_{I_t} \mid \mathcal{S}_t) = H_\theta(X_{i_1}, \ldots, X_{i_k} \mid \mathcal{S}_t)$, and the planner can be viewed as optimizer to choosing $I_t$ that reduce this joint entropy over the course of sampling.

In practice, the denoiser $\mathcal{D}_\theta$ outputs a categorical distribution for each masked position conditioned on $\mathcal{S}_t$, which gives direct access to the per-token conditional entropies $H_\theta(X_{i_j} \mid \mathcal{S}_t)$ but not to the full joint distribution over $(X_{i_1}, \ldots, X_{i_k})$. The exact joint entropy $H_\theta(X_{i_1}, \ldots, X_{i_k} \mid \mathcal{S}_t)$ under the sampling process is therefore not tractable. Using the chain rule for entropy and the fact that conditioning reduces entropy (Cover & Thomas, 2006, Theorems. 2.5.1, 2.6.5), we obtain

$$H_\theta(X_{i_1}, \ldots, X_{i_k} \mid \mathcal{S}_t) \ \leq \ \sum_{j=1}^{k} H_\theta(X_{i_j} \mid \mathcal{S}_t), \tag{3}$$

with equality if the positions are conditionally independent given $\mathcal{S}_t$. The RHS of this inequality, $\sum_{j=1}^{k} H_\theta(X_{i_j} \mid \mathcal{S}_t)$, is thus used as a tractable surrogate objective for planner design. The following sections compare several planner strategies, including DUS, and analyze their behavior in terms of the resulting block conditional entropy across unmasking iterations.

### 3.4 SELF-PLANNERS GUIDED BY DENOISER CONFIDENCE

Self-planners use the denoiser's output distribution to score each masked position and decide which indices to unmask. At iteration $t$, each masked index $i$ receives a score $s(i \mid \mathcal{S}_t)$ computed from $p_\theta(X_i \mid \mathcal{S}_t)$, and the planner selects either the top-$k$ positions or an adaptive number of positions whose scores exceed a threshold (Wu et al., 2025; Ben-Hamu et al., 2025). A probability-based self-planner ranks positions by confidence, e.g. $s(i \mid \mathcal{S}_t) = \max_x p_\theta(X_i \mid \mathcal{S}_t)$, and selects the top-$k$ masked tokens with the largest scores. An entropy-based self-planner instead uses the individual conditional entropy, $s(i \mid \mathcal{S}_t) = -H_\theta(X_i \mid \mathcal{S}_t)$, so that tokens with lower uncertainty receive higher scores.

In the notation of Section 3.3, both types of self-planner ultimately choose an index set $I_t = \{i_1, \ldots, i_k\}$ based solely on per-token statistics from the denoiser. Their selection rules implicitly minimize the surrogate

$$\sum_{j=1}^{k} H_\theta(X_{i_j} \mid \mathcal{S}_t), \tag{4}$$

which is the upper bound on the true joint conditional entropy $H_\theta(X_{i_1}, \ldots, X_{i_k} \mid \mathcal{S}_t)$ introduced earlier, and therefore do not explicitly control the dependencies between the selected positions.

## 3.5 DUS as Predefined Planner

DUS is a planner that, for each block of size $B$ and base $a > 1$, fixes in advance which positions will be unmasked at each of $R$ iterations, independently of the model outputs. Inside a block, positions are indexed by $\{1, \ldots, B\}$, and the number of iterations and step sizes are defined as

$$R = \lceil \log_a B \rceil, \quad s_t = \lfloor \tfrac{B}{a^t} \rfloor, \quad t = 1, \ldots, R. \tag{5}$$

The planner output $\mathcal{P}_t$ at iteration $t$ and the incremental unmasked set $\mathcal{U}_t$ are defined as,

$$\mathcal{P}_t = \big\{ k \in \{1, \ldots, B\} \setminus \mathcal{U}_{t-1} \mid (k-1) \bmod s_t = 0 \big\}, \quad \mathcal{U}_0 = \emptyset, \quad \mathcal{U}_t = \mathcal{U}_{t-1} \cup \mathcal{P}_t. \tag{6}$$

If $|\mathcal{P}_R| < |\mathcal{P}_{R-1}|$, then $\mathcal{P}_R$ is merged into $\mathcal{P}_{R-1}$ to balance coverage. This schedule completes a block in $R \approx \lceil \log_a B \rceil$ iterations, where the values $s_t$ make early iterations unmask a few widely spaced positions and later iterations fill in the remaining gaps at finer resolution. For example, let $B = 8$ and $a = 2$. Then $R = 3$, $s_1 = 4$, $s_2 = 2$, $s_3 = 1$, and

$$\mathcal{P}_1 = \{ k \mid (k-1) \bmod 4 = 0 \} = \{1, 5\},$$
$$\mathcal{P}_2 = \{ k \notin \{1, 5\} \mid (k-1) \bmod 2 = 0 \} = \{3, 7\},$$
$$\mathcal{P}_3 = \{ k \notin \{1, 3, 5, 7\} \mid (k-1) \bmod 1 = 0 \} = \{2, 4, 6, 8\}.$$

## 3.6 Theoretical Analysis of DUS

Under the VLMC with bounded order and fast-mixing property, the next lemma shows that if the indices selected in a single iteration are sufficiently well separated, then the joint conditional entropy of these tokens is $\varepsilon$ close to the sum of their individual conditional entropies.

**Lemma 1** *Assume that $(X_1, \ldots, X_G)$ is a stationary, ergodic VLMC with finite order $L$ with fast-mixing property. Let $1 \le i_1 < \cdots < i_k \le B$ be indices such that the pairwise distances satisfy $|i_m - i_n| \ge d$ for all $m \ne n$ and $d \le L$. Then, for any $\varepsilon > 0$, there exists a distance threshold $D_\varepsilon$ such that for all $d \ge D_\varepsilon$,*

$$H_\theta\big(X_{i_1}, \ldots, X_{i_k} \mid \mathcal{S}_t\big) \ge \sum_{j=1}^{k} H_\theta\big(X_{i_j} \mid \mathcal{S}_t\big) - \varepsilon. \tag{7}$$

The proof of Lemma 1 relies on the fast-mixing property of the underlying VLMC, which implies that the mutual information (MI) between a token and any finite block of sufficiently distant tokens decays rapidly with their separation. The following lemma formalizes this decay. Restated lemma and proof are given in Appendix B.2.

**Lemma 2 (MI decays under fast mixing)** *Let $(X_t)_{t \in \mathbb{Z}}$ be a stationary, ergodic VLMC with finite order $L < \infty$, fast-mixing property on a finite alphabet (Assumption 1 in Appendix B.1). Then there exist constants $C < \infty$ and $\rho \in (0, 1)$ such that for any index $i$, spacing $1 \le d \le L$, and any finite $M \ge 0$,*

$$I_\theta\big(X_i; X_{i+d}, X_{i+2d}, \ldots, X_{i+(M+1)d}\big) \le C\rho^d. \tag{8}$$

*A proof of equation 8 is given in Appendix B.1.*

In the context of DUS, the indices in $\mathcal{P}_t$ within a fixed iteration $t$ satisfy a minimum separation $s_t$ in index space (Section 3.5). Combined with Lemma 2, this spacing implies that the MI between selected positions can be made uniformly small by choosing sufficiently large $s_t$, which in turn yields the joint-entropy bound in Lemma 1 for the set $\mathcal{P}_t$.

In summary, DUS is designed to minimize the gap between the tractable sum of per-token conditional entropies and the intractable joint conditional entropy. In contrast, generic self-planners may select tightly clustered tokens for which the objective can be a loose upper bound on the true joint entropy. Three core features are integrated in the planner:

1. **Maintained spacing across iterations.** By preserving the same sparse dilation at every round, tokens distant is guaranteed - and thus the bound in Lemma 1 holds throughout all $R$ iterations, whereas schedulers that repeatedly cluster adjacent tokens can incur larger MI penalties and make the aforementioned entropy gap larger.

Table 1: Math (GSM8K, MATH500) and code (Humaneval, MBPP) benchmarks for self-confidence (Conf.) and DUS (ours). Tasks reported accuracy (%) for math and pass@1 for code, at block sizes $B = \{8, 16, 32, 64\}$, with corresponding speedup factor ($\times$). Model names: B=Base, I=Instruct. Diffucoder is tested only on code benchmarks; Humaneval, and MBPP. **Bold** marks better planner scores. $*$ denotes our own reruns for - token-by-token MDLM baselines, [†]Llama-3-8B, and [‡]Qwen-3-8B.

| | Model | $\times 1$ | B=8 $\times 2.7$ | | B=16 $\times 4$ | | B=32 $\times 6.4$ | | B=64 $\times 10.7$ | | AR Baseline |
|---|---|---|---|---|---|---|---|---|---|---|---|
| | | Conf. | Conf. | DUS | Conf. | DUS | Conf. | DUS | Conf. | DUS | L[†] / Q[‡] |
| GSM8K | LLaDA-B | 72.63* | 59.29 | **63.08** | 51.23 | **59.51** | 29.04 | **49.36** | 8.04 | **35.18** | |
| | LLaDA-I | 80.29* | 69.22 | **73.24** | 61.41 | **70.66** | 38.74 | **65.73** | 18.73 | **57.09** | 49.81* / 88.55* |
| | Dream-I | 77.10* | 61.64 | **65.28** | 53.22 | **56.63** | 27.60 | **44.66** | 17.89 | **32.07** | |
| MATH500 | LLaDA-B | 24.00* | 16.6 | **21.4** | 11.2 | **19.2** | 6.0 | **13.6** | 2.6 | **10.2** | |
| | LLaDA-I | 28.80* | 21.4 | **23.8** | 15.4 | **22.8** | 10.8 | **19.2** | 8.0 | **14.8** | 15.23* / 50.20* |
| | Dream-I | 37.00* | 22.4 | **27.0** | 15.4 | **19.8** | 7.2 | **13.2** | 4.0 | **11.6** | |
| Humaneval | LLaDA-B | 34.76* | 15.85 | **25.61** | 12.8 | **19.51** | 4.88 | **14.02** | 4.88 | **6.71** | |
| | LLaDA-I | 39.02* | 21.95 | **28.05** | 14.02 | **23.17** | 9.76 | **10.37** | 10.98 | **11.59** | |
| | Dream-I | 57.90 | 8.54 | **14.63** | 5.49 | **11.59** | **6.71** | 6.71 | 6.10 | **9.15** | 36.59* / 61.59* |
| | DiffuCoder-B | 67.10 | 17.07 | **28.66** | 6.71 | **38.41** | 2.44 | **21.95** | 0.61 | **6.10** | |
| | DiffuCoder-I | 72.00 | 7.93 | **22.56** | 14.02 | **20.12** | **13.41** | 12.80 | 11.59 | 8.54 | |
| MBPP | LLaDA-B | 38.0* | 19.8 | **30.4** | 12.8 | **31.6** | 8.2 | **22.6** | 3.4 | **14.4** | |
| | LLaDA-I | 39.4* | 25.4 | **33.6** | 17.6 | **31.8** | 14.0 | **23.2** | 11.4 | **18.6** | |
| | Dream-I | 56.2 | 32.8 | **45.0** | 23.8 | **40.8** | 16.4 | **26.6** | 11.8 | **22.2** | 48.4* / 65.4* |
| | DiffuCoder-B | 74.2 | 29.2 | **48.6** | 17.4 | **43.0** | 10.2 | **27.4** | 3.4 | **17.2** | |
| | DiffuCoder-I | 75.1 | 31.8 | **46.4** | 25.6 | **43.6** | 21.0 | **26.6** | 13.0 | **18.2** | |

2. **Contextual conditioning.** By revealing tokens spread across the sequence early, later predictions are conditioned on richer local context from multiple regions of the block. This tends to reduce the remaining conditional entropies and increases the effective context available to the denoiser compared to planners that focus on a contiguous region.

3. **Skip mechanism.** As an interleavable feature atop the deterministic schedule, tokens whose denoiser signal (e.g., low confidence or low negative entropy) falls below a threshold are deferred to the next iteration $\mathcal{P}_{t+1}$. This allows DUS to preserve its fixed dilation pattern while still adapting to model uncertainty and avoiding the premature exposure of highly uncertain tokens.

Under the fast-mixing VLMC, the analysis above suggests that DUS achieve lower cumulative conditional entropy than denoiser-guided planners in parallel unmasking. In settings where long-range dependencies are especially strong (e.g., rhyme schemes or variables in code), any parallel unmasking scheme, including DUS, can still unmask such tokens in parallel and incur larger entropy gaps; nevertheless, because DUS keeps updates spread out and many tasks admit multiple valid completions (e.g., different correct programs, proofs, or phrasings), later iterations can still repair local inconsistencies around early parallel decisions by adjusting neighboring tokens.

## 3.7 EXPERIMENTAL SETUP

For each model and dataset, decoding is applied with a semi-AR masked diffusion denoising process for different block sizes $B \in \{8, 16, 32, 64\}$. Decoding proceeds in $n_{\text{blocks}} = G/B$ rounds, in each of which the model predicts all currently masked tokens before moving to the next block. The Number of Function Evaluations (NFE) of a block is $\text{NFE}_{\text{block}} = \log_2 B$ on average - due to the nature of our DUS planner. Thus, $k$, the number of unmasked tokens in parallel in one iteration is set to $k = B/\log_2 B$ on average. The total number of denoiser calls - total NFE, is

$$\text{NFE} = n_{\text{blocks}}\text{NFE}_{\text{block}} = \frac{G}{B}\log_2 B, \qquad (9)$$

hence larger blocks results in with more parallelism against fewer diffusion evaluations. Section 4.3 investigate further the effect on performance of fixed $k$ versus incremental $k$ (as in DUS), for self-confidence and random planners. Beside this particular experiment, all experiment's other planner but DUS feature fixed $k$.

Table 2: General knowledge (BBH, MMLU-pro) benchmarks for self-confidence (Conf.) and DUS (ours). Both are few-shot, COT, multiple-choice datasets on general topics from various fields. Tasks report accuracy (%) for token-by-token ($\times 1$) and block sizes $B = \{8, 16, 32, 64\}$, with corresponding speedup factor ($\times$). Model names: B=Base, I=Instruct. For token-by-token baselines, $*$ denotes our own reruns. **Bold** marks better planner scores. $^{\dagger}$Llama-3-8B; $^{\ddagger}$Qwen-3-8B.

| | Model | $\times 1$ | B=8 $\times 2.7$ | | B=16 $\times 4$ | | B=32 $\times 6.4$ | | B=64 $\times 10.7$ | | AR Baseline |
|---|---|---|---|---|---|---|---|---|---|---|---|
| | | Conf. | Conf. | DUS | Conf. | DUS | Conf. | DUS | Conf. | DUS | $L^{\dagger}$ / $Q^{\ddagger}$ |
| BBH | LLaDA-B | 44.26$^*$ | 41.67 | **43.33** | 40.93 | **41.48** | 35.93 | **40.56** | 26.67 | **38.52** | |
| | LLaDA-I | 53.89$^*$ | 51.48 | **51.67** | 47.59 | **50.37** | 44.26 | **50.93** | 41.67 | **50.37** | 52.59$^*$ / 59.63$^*$ |
| | Dream-I | 58.15$^*$ | 55.93 | **54.81** | 50.93 | **53.52** | 47.04 | **50.56** | 36.30 | **37.41** | |
| MMLU-Pro | LLaDA-B | 39.82$^*$ | 36.07 | **41.96** | 32.14 | **37.32** | 24.11 | **32.50** | 16.07 | **30.36** | |
| | LLaDA-I | 40.89$^*$ | 34.46 | **35.89** | 25.71 | **34.64** | 31.07 | **34.64** | 16.07 | **32.68** | 37.86$^*$ / 52.68$^*$ |
| | Dream-I | 50.89$^*$ | 46.43 | **48.04** | 36.96 | **47.32** | 27.50 | **48.39** | 13.75 | **39.64** | |

We evaluate five masked diffusion LLMs are evaluated. (1) LLaDA-Base-8B, (2) LLaDA-Instruct-8B (Nie et al., 2025b) - an 8B parameter mask-diffusion transformer pre-trained on a large mixed text + code corpus, and it's instruction-tuned version. (3) Dream-Instruct-7B (Ye et al., 2025b) - a 7B parameter instruction-tuned mask-diffusion model, supervised fine-tuned based on the base version. To streamline our analysis, we focus on the instruction-tuned variant and do not report results for the pre-trained Dream-Base model, whose out-of-the-box performance for our experiments is appreciably lower than that of its fine-tuned version. (4) DiffuCoder-Base-7B, (5) DiffuCoder-Instruct-7B (Gong et al., 2025) - a 7B parameter mask-diffusion transformer trained exclusively on 130B tokens of code with AR-initialization, and it's instruction fine-tuned model. Diffucoder authors claim that their results are given without semi-AR protocol, hence for their model without DUS semi-AR inference is off. Two planners are used:

1. **Self-confidence (baseline).** at each block the model's top-$k$ confident tokens are unmasked at each diffusion reverse iteration, while the others are masked. $k$ is set to a fixed value that is dependent on block size of the semi-AR process, $k = \log_2 B$, unless stated otherwise. LLaDA and Dream models use maximum probability as their confidence while Diffucoder uses entropy (as in their original work).

2. **DUS (ours).** for each block length $B$ the DUS is applied (as defined in Section 3.5), that unmasks on average $k = \log_2 B$ tokens in denoiser iteration, across a block.

Experiments report block size $B$ and relative inference speedup, $B/\log_2 B$ (originates from Equation equation 9), calculated as the ratio of token-by-token total NFE to the experiment's total NFE; both planners use the same $B$ and NFE budget, and their final task scores are compared accordingly. A GSM8K problem, for $B = 32$, is visualized in Figure 1b and Figure 1c for DUS and self-confidence respectively.

# 4 EXPERIMENTS

This section evaluates the generative and downstream capabilities of our new planner for MDLMs on various families of tasks: mathematical reasoning, code generation and general knoeledge COT multiple-choice questions. Three pretrained diffusion-based LLM - LLada, Dream, and Diffucoder - under unified benchmarking protocols. Detailed information about datasets and experiments settings presented in Appendix A.1 The following subsections first study math and coding benchmarks (Section 4.1), then general knowledge multiple choice reasoning (Section 4.2), and finally ablations on planner parallelism and NFE (Section 4.3).

## 4.1 MATH AND CODING EXPERIMENTS

Models are evaluated on GSM8K (Cobbe et al., 2021), MATH500 (Lightman et al., 2023), HumanEval Chen et al. (2021), and MBPP Austin et al. (2021) to assess performance trade-offs between inference speed and accuracy under semi-AR masked diffusion. Figure 1a and 2 plot accuracy vs. speedup for $2.7\times$, $4\times$, $6.4\times$, and $10.7\times$ (block sizes $B \in \{8, 16, 32, 64\}$). Smaller blocks yield

Table 3: Planners parallelism ablation on GSM8K (300 samples, $G = 256$), comparing self-confidence (Conf.) vs. random planners under two unmasking schedules: fixed-$k$ and dilated-incremental (Inc., as in DUS). Accuracy (%) is shown for both base and instruct LLaDA models at $B = \{16, 32\}$. Best scores are **bold**, second-best are underlined.

| Model | Block | Conf. | | Random | | DUS |
|---|---|---|---|---|---|---|
| | | fixed | inc. | fixed | inc. | inc. |
| Base | 16 | 55.00 | 49.33 | 24.67 | 43.67 | **61.33** |
| | 32 | 30.67 | 25.00 | 10.00 | 39.00 | **48.33** |
| Instruct | 16 | 62.33 | 47.67 | 40.00 | 61.33 | **71.67** |
| | 32 | 39.67 | 23.67 | 23.33 | 53.67 | **66.67** |

higher accuracy at the cost of more denoiser rounds (higher NFE), while larger blocks enable up to $10\times$ fewer iterations but lower end accuracy. The DUS planner consistently improves up to 27% over the self-confidence baseline at the same NFE budget. GSM8K shows a steady decline in scores as inference speed increases, whereas the code benchmarks (HumanEval and MBPP) exhibit less smooth trends-likely reflecting their already low performance (under 10 %) at higher speedups. Nonetheless, DUS delivers more consistent improvements across all settings.

Table 1 presents extensive results on all of the 4 math and code benchmarks. Consistently, DUS improves or matches the results of the baseline self-confidence planner, as it can achieve up to 40% improvement on math benchmarks and up to 20% on code benchmarks, while using less NFE compared to naive token-by-token inference. For reference, it is also reports token-by-token self-confidence baselines ($\times 1$ speedup), to compare speedup-accuracy trade-offs.

Moreover, it can be observed that particularly on code benchmarks, LLaDA and Diffucoder base models outperform their instruct model while using DUS as their planner, compared to the superiority of instruction tuned models with self-confidence planner.

## 4.2 General Knowledge Experiments

To further demonstrate DUS's superiority under accelerated inference budgets, experiments were conducted on the BBH (Suzgun et al., 2022) and MMLU-Pro (Wang et al., 2024) multiple-choice reasoning benchmarks using a few-shot chain-of-thought protocol. Table 2 reports token-by-token self-confidence baselines ($\times 1$) and semi-AR block diffusion results for $B \in \{8, 16, 32, 64\}$, including LLaDA-Base, LLaDA-Instruct, and Dream-Instruct. Although absolute gains are smaller than in the math and code experiments, DUS consistently outperforms the self-confidence planner at the same speedups: BBH accuracy improves by 1-5%, and MMLU-Pro by 5-9%. These results confirm that DUS yields higher-quality reasoning across diverse question formats under constrained inference budgets.

## 4.3 Planners Parallelism Performance

**Power of spreading.** To test whether DUS's improvements come from its incremental unmasking schedule rather than simply unmasking more tokens per step, the number of tokens revealed per iteration is varied under two schedules: (i) *fixed-k*, which unmasks a constant $k = \log_2 B$ tokens at every step, and (ii) *dilated-incremental*, which gradually increases the unmasking group so that the average remains $k = \log_2 B$ but early iterations reveal fewer tokens and later iterations more, aligning the concepts of cosine scheduling for discrete diffusion (Shi et al., 2025), but matching the amount pattern of DUS. Each schedule is applied with both a self-confidence planner and a random planner and report GSM8K accuracy on the first 300 samples for LLaDA-Base and LLaDA-Instruct at block sizes $B = 16, 32$ in Table 3.

Across all configurations, DUS attains the highest score. Under self-confidence, fixed-$k$ slightly outperforms dilated-incremental, suggesting that confidence-guided selection benefits from uniform parallelism. By contrast, the random planner performs poorly with fixed-$k$ but, when paired with the dilated-incremental schedule, recovers most of its accuracy loss and can even surpass fixed-$k$ self-confidence variants. These ablations confirm that DUS's gains do not arise merely from unmasking

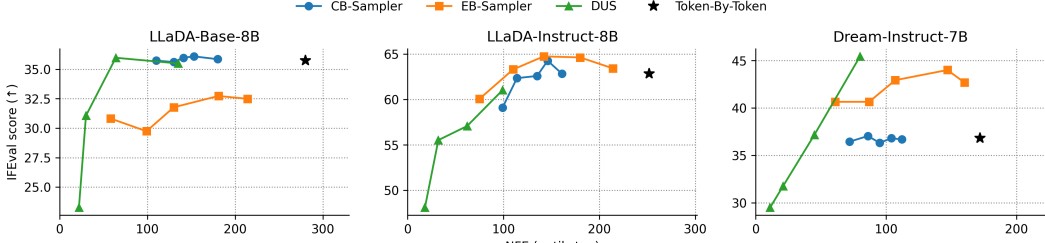

Figure 3: IFEval score versus NFE (for early termination) for CB-Sampler (blue), EB-Sampler (orange), and DUS (green) on LLaDA-Base-8B, LLaDA-Instruct-8B, and Dream-Instruct-7B. CB varies the confidence temperature $\tau = \{0.5, 0.6, 0.7, 0.9\}$, EB varies the entropy bound $\gamma = \{0.01, 0.1, 1, 2, 4\}$, and DUS varies the block size $B = \{8, 16, 32, 64\}$. For comparison, AR model, Llama-3-8B achieves score of 15.23 with 356 NFEs on average, and Qwen-3-8B achieves 41.73 with 577 NFEs on average. $\star$ denotes token-by-token greedy sampler. Higher accuracy (%) and lower NFE are better.

more tokens per step: even when the total parallelism budget is held fixed and shared with a random planner, the spacing-aware schedule is crucial for preserving accuracy under aggressive parallel decoding.

**Guaranteed speedups.** IFEval (Zhou et al., 2023) is an instruction following benchmark that evaluates the accuracy of LLMs to follow and execute them. Figure 3 compares CB-Sampler, EB-Sampler, and DUS on IFEval for LLaDA-Base-8B, LLaDA-Instruct-8B, and Dream-Instruct-7B. Each point reports IFEval score versus the empirical number of denoising steps until an `<EOS>` token appears (NFE until early stop) on a 1024 generation length, with comparison to greedy token-by-token decoding. CB and EB vary their confidence and entropy thresholds, and DUS varies the block size $B$.

On LLaDA-Base, DUS with $B = 16$ matches the token-by-token score (35.97 vs. 35.73) while reducing NFE from about 280 to 64 steps, a speedup of more than $4\times$, and remains below its analytic upper bound $(G/B) \log B$. On LLaDA-Instruct, EB with an intermediate entropy bound ($\gamma = 1$) achieves the best score, 64.75 at 142 NFEs, modestly improving over the greedy baseline, while DUS achieves slightly lower scores, but for nearly $3\times$ faster then the last. On Dream-Instruct-7B, DUS with $B = 8$ attains the highest score (45.44) at only 80 NFEs, compared to EB's best score of 44.00 at 147 NFEs and CB's 37.05 at 86 NFEs.

These results show that parallel unmasking can match or exceed token-by-token decoding while also reducing the NFE substantially. While the effective speedups of CB and EB samplers remain stochastic, since they depend on the denoiser's output and threshold tuning, DUS delivers predictable speedups, often even better than the theoretical $(G/B) \log B$ NFE budget. Overall, DUS provides a good trade-off between accuracy and inference time.

## 5 Conclusions

We introduced DUS, a purely inference-time, model-free agnostic planner for MDLMs. In extensive experiments on diffusion LLMs, DUS consistently outperforms the traditional denoiser-confidence planner, improving downstream task accuracy by up to 27% on challenging math and code benchmarks, while simultaneously reducing the number of denoising iterations by an order of magnitude. Unlike typical speed-quality trade-offs, our method both accelerates inference and enhances output quality. By unlocking the parallelism inherent in diffusion decoding without any modifications to model architecture or training, DUS demonstrates a new path toward diffusion-based LMs that surpass AR approaches in both efficiency and performance, and inspires the design of future inference-only planners to fully exploit this potential.

**Use of LLMs.** A large language model was used solely for minor editing of draft text. The model was not used for ideation, experimental design, implementation, analysis, data generation, or to produce technical content. All scientific claims and results were produced and verified by the authors.

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

# A  APPENDIX

## A.1  DATASETS AND SETTINGS

All experiments were conducted on NVIDIA Tesla V100 GPUs. Evaluation was performed using the *Language Model Evaluation Harness* repository (https://github.com/EleutherAI/lm-eval-harness) and our code is based on the LLaDA repository https://github.com/ML-GSAI/LLaDA. Average NFEs per block (and overall for each generation length) are reported to ensure a fair comparison across methods.

For the standard benchmarks GSM8K (Cobbe et al., 2021), MBPP (Austin et al., 2021), and HumanEval (Chen et al., 2021), evaluation is conducted on the full datasets (1,319, 500, and 164 samples, respectively), with GSM8K parallelism ablations limited to the first 300 samples. To facilitate evaluation on MATH500 (Lightman et al., 2023), a new task class is implemented for the 500 cherry-picked samples from the MATH dataset (Hendrycks et al., 2021). Finally, to assess robustness on reasoning and professional-knowledge benchmarks, subsets are sampled from BBH (Suzgun et al., 2022) (20 examples from each of its 27 subgroups, 540 total) and MMLU-Pro (Wang et al., 2024) (40 examples from each of its 14 subgroups, 560 total). Datasets' generation length $G$ is set to 256, except for the coding datasets HumanEval and MBPP, where $G = 512$. For IFEval (Zhou et al., 2023), which is used in the planner parallelism experiments, the generation length is set to $G = 1024$.

All evaluations use a few-shot, COT prompting framework: GSM8K and MBPP with 4-shot contexts, HumanEval with 0-shot, MATH500 with 4-shot, BBH with 3-shot, and MMLU-Pro with 5-shot.

Early stop for generation is implemented for all planners, which ends generation if the [EOS] token is unmasked and all previous tokens are unmasked too, since generating text after this token would produce content unrelated to the answer.

Table 4 summarizes the notation and formulas used throughout this paper. In most experiments, the unmasking parameter is set as $k = \log_2 B$ unless otherwise specified.

| Formula | Definition |
|---------|------------|
| $G$ | Generation length. |
| $B$ | Block size. |
| $k$ | Average number of unmasked tokens per iteration. |
| $n_{\text{blocks}} = \dfrac{G}{B}$ | Number of blocks. |
| $\text{NFE}_{\text{block}} = \dfrac{B}{k}$ | Number of function evaluations per block. |
| $\text{NFE} = n_{\text{blocks}} \cdot \text{NFE}_{\text{block}}$ | Total number of function evaluations. |
| $\dfrac{\text{NFE}}{\text{NFE}_{\text{block}}} = \dfrac{B}{k}$ | Speedup factor compared to token-by-token inference. |

Table 4: Inference speedup conversion table.

Table 5: Speedup comparison on GSM8K (300 samples, $G = 256$) using DUS under an $8\times$ inference budget (total NFE $= 32$). Block sizes $B \in \{8, 16, 32\}$ correspond to average NFEs per block of $\{1, 2, 4\}$. Results for both base and instruct LLaDA models; best scores are **bold** and second-best are underlined.

| Block Size | 8 | 16 | 32 |
|------------|-----|-----|-----|
| **Avg NFEs@Block** | 1 | 2 | 4 |
| **Model** | Score (%, ↑) | | |
| Base | 13.33 | 11.00 | **32.33** |
| Instruct | 12.00 | 16.00 | **56.67** |

## A.2 BLOCK-SIZE EFFECT ON SPEEDUPS

In this experiment, the effect of block size $B$ on generation accuracy was evaluated under a fixed total NFEs, corresponding to an $8\times$ speedup relative to token-by-token decoding. The DUS was configured to begin at a higher iteration $t_0 > 1$, resulting in larger unmasking group sizes $k$ per step (cf. equation 5). Block sizes $B = \{8, 16, 32\}$, corresponding to $\text{NFE}_{\text{block}} = \{1, 2, 4\}$, were tested on GSM8K (first 300 samples). Table 5 reports task accuracy (%) for LLaDA-Base and LLaDA-Instruct, revealing a monotonic increase in performance with $B$: the Base model rose from 13.33% at $B = 8$ to 32.33% at $B = 32$ ($\approx 2.4\times$ accuracy improvement), while the Instruct variant climbed from 12.00% to 56.67% ($\approx 4.7\times$ accuracy improvement).

These gains are attributed to the fact that larger block sizes, when combined with DUS, spread the initially predicted tokens farther apart. This spatial separation reduces MI among unmasked tokens in early iterations - consistent with our analysis in Lemma 2 - and allows subsequent iterations to fill in the gaps and more effectively correct existing errors introduced by coarse-grained parallelism. Tuning $B$ thus offers an additional lever to boost output quality without raising the compute budget, though excessively large $B$ may force the model to predict tokens with insufficient nearby context, which can exceed the model capabilities.

## B THEORY GUARANTEES

### B.1 MI DECAY UNDER FAST MIXING

This appendix justify Lemma 2, which bounds the MI between a token and a set of sparsely sampled past tokens under the fast-mixing VLMC assumption.

**Assumption 1 (Fast mixing VLMC)** *Let $(X_i)_{i \in \mathbb{Z}}$ be a VLMC with finite maximal context $L < \infty$ on a finite alphabet. Assuming the induced Markov chain on contexts of length $L$ is irreducible and aperiodic, which implies a unique stationary distribution and geometric mixing.*

**Lemma 2 (MI decay under fast mixing) (restated).** *Under Assumption 1, the MI decays geometrically with the distance between tokens for $k \leq L$,*

$$I(X_i; X_{i+k}, X_{i+2k}, \ldots, X_{i+(M+1)k}) \leq \mathcal{O}(\rho^k) \to 0 \quad \text{as } k \to \infty, \tag{10}$$

*for some $0 < \rho < 1$ and finite $M$.*

Note that the distance between $X_i$ and any symbol in the future block is at least $k$.

**Proof:** Define contexts of length $L$ as $C_i = (X_i, X_{i+1}, \ldots, X_{i+(L-1)}) \in \mathcal{C}$. The sequence $(C_i)$ forms a finite-state Markov chain which is irreducible and aperiodic, implying geometric mixing. Observe that a block of $M + 1$ future symbols is a deterministic function of the future contexts,

$$(X_{i+k}, X_{i+2k}, \ldots, X_{i+(M+1)k}) = f(C_{i+k}, C_{i+2k}, \ldots, C_{i+(M+1)k}). \tag{11}$$

The data processing inequality (DPI) states that if $Y = f(Z)$ is a function of $Z$, then $I(X;Y) \leq I(X;Z)$. Letting $Y = (X_{i+k}, X_{i+2k}, \ldots, X_{i+(M+1)k})$ and $Z = (C_{i+k}, C_{i+2k}, \ldots, C_{i+(M+1)k})$, it follows that

$$I(X_i; X_{i+k}, X_{i+2k}, \ldots, X_{i+(M+1)k}) \leq I(X_i; C_{i+k}, C_{i+2k}, \ldots, C_{i+(M+1)k}). \tag{12}$$

Since $X_i$ is a function of the current context $C_i$, applying DPI again yields

$$I(X_i; C_{i+k}, C_{i+2k}, \ldots, C_{i+(M+1)k}) \leq I(C_i; C_{i+k}, C_{i+2k}, \ldots, C_{i+(M+1)k}). \tag{13}$$

Thus, it suffices to show that the MI between the current context $C_i$ and a block of future contexts decays with the separation $k$.

Since $(C_i)$ is finite-state, irreducible, and aperiodic (because $k \leq L$, there exists $\alpha < \infty$ and $0 < \rho < 1$ such that for any state $c$ the total variation (TV) between the stationary and conditioned on past probabilities are bounded by exponentially decays factor dependent on $k$ (Bradley, 2005),

$$\|P(C_{i+k} \mid C_i = c) - \pi\|_{TV} \leq \alpha \rho^k. \tag{14}$$

For a block of $M + 1$ future contexts, standard finite-state Markov chain bounds imply

$$\|P(C_{i+k}, C_{i+2k}, \ldots, C_{i+(M+1)k} \mid C_i = c) - \pi^{\otimes(M+1)}\|_{TV} \leq (M+1)\alpha \rho^k. \tag{15}$$

The MI can be bounded in terms of TV distance (Zhang, 2007). In particular, if $\|P_{AB} - P_A \otimes P_B\|_{TV} \leq \delta$, then $I(A;B) \leq \beta \delta \log(1/\min(P_B))$ for some constant $\beta$. Plugging it in equation 14 gives,

$$I(C_i; C_{i+k}, C_{i+2k}, \ldots, C_{i+(M+1)k}) \leq \beta(M+1)\alpha \rho^k \to 0 \quad \text{as } k \to \infty. \tag{16}$$

Combining these inequalities results in

$$I(X_i; X_{i+k}, X_{i+2k}, \ldots, X_{i+(M+1)k}) \leq I(C_i; C_{i+k}, C_{i+2k}, \ldots, C_{i+(M+1)k}) \to 0. \tag{17}$$

Therefore, the MI between $X_i$ and any finite block of sufficiently far future symbols vanishes with a geometric rate in $k$. $\qquad\square$

### B.2 Lemma 1 - Restate and proof

**Lemma 1 (restated).** *Assume that $(X_1, \ldots, X_G)$ is a stationary, ergodic VLMC with finite order $L$ with fast-mixing property. Let $1 \leq i_1 < \cdots < i_k \leq B$ be indices such that the pairwise distances satisfy $|i_m - i_n| \geq d$ for all $m \neq n$ and $d \leq L$. Then, for any $\varepsilon > 0$, there exists a distance threshold $D_\varepsilon$ such that for all $d \geq D_\varepsilon$,*

$$H_\theta(X_{i_1}, \ldots, X_{i_k} \mid \mathcal{S}_t) \geq \sum_{j=1}^{k} H_\theta(X_{i_j} \mid \mathcal{S}_t) - \varepsilon. \tag{18}$$

**Proof of Lemma 1:** Fix $\varepsilon > 0$. By Lemma 2, there exists $D_\varepsilon$ such that for any pair of indices $(i_1, \ldots, i_k)$ with $|i_m - i_n| \geq D_\varepsilon$ for all $m \neq n$, and any realization of $\mathcal{S}_t$,

$$I_\theta(X_{i_k}; X_{i_1}, \ldots, X_{i_{k-1}} \mid \mathcal{S}_t) \leq \frac{\varepsilon}{(k-1)^2}. \tag{19}$$

Assume $d \geq D_\varepsilon$, so that equation 19 holds for all $m \neq n$.

Without loss of generality, let $i_1 < \cdots < i_k$ be the indices of the $k$ tokens chosen in this iteration, each pair separated by at least $D_\varepsilon$. The joint conditional entropy decomposes as

$$H_\theta(X_{i_1}, \ldots, X_{i_k} \mid \mathcal{S}_t) = \sum_{j=1}^{k} H_\theta(X_{i_j} \mid X_{i_1}, \ldots, X_{i_{j-1}}, \mathcal{S}_t)$$

$$= H_\theta(X_{i_1} \mid \mathcal{S}_t) + \sum_{j=2}^{k} \left[ H_\theta(X_{i_j} \mid \mathcal{S}_t) - I_\theta(X_{i_j}; X_{i_1}, \ldots, X_{i_{j-1}} \mid \mathcal{S}_t) \right].$$

By equation 19,

$$I_\theta(X_{i_j}; X_{i_1}, \ldots, X_{i_{j-1}} \mid \mathcal{S}_t) \leq (j-1) \cdot \frac{\varepsilon}{(k-1)^2} \leq \frac{\varepsilon}{k-1}. \tag{20}$$

Plugging this into the entropy decomposition above, with $k-1$ bounded terms of MI gives

$$H_\theta(X_{i_1}, \ldots, X_{i_k} \mid \mathcal{S}_t) \geq \sum_{j=1}^{k} H_\theta(X_{i_j} \mid \mathcal{S}_t) - \varepsilon, \tag{21}$$

which is exactly the claimed inequality equation 7. $\qquad\square$

## C  QUALITATIVE GENERATION TRAJECTORIES

This section qualitatively visualize the generation process of LLaDA-8B on several benchmarks under two decoding strategies: our Dilated Unmasking Scheduler (DUS) and a fixed block scheduler. For each dataset, a representative example selected, the full generated answer for four decoding configurations are presented (DUS with two block sizes and fixed scheduling with self-confidence with two block sizes). Tokens are colored according to the diffusion step at which they are first unmasked: early steps appear in blue, intermediate steps in green, and late steps in yellow to red. All prompts are given without their few-shots. The prompt length is reported in tokens for each example.

The background color of each token indicates when it was unmasked during generation:

Early (Stage 1) $\rightarrow$ Quarter $\rightarrow$ Middle $\rightarrow$ Three-Quarter $\rightarrow$ Late (Final Stage)

## C.1 MATH500 - LLADA-8B-INSTRUCT

QUESTION (NO FEW-SHOTS)

PROMPT LENGTH: 504 TOKENS

> Problem: A regular pentagon is rotated counterclockwise about its center. What is the minimum number of degrees it must be rotated until it coincides with its original position?

**DUS B=32** (Steps: 11, Generation length: 81)

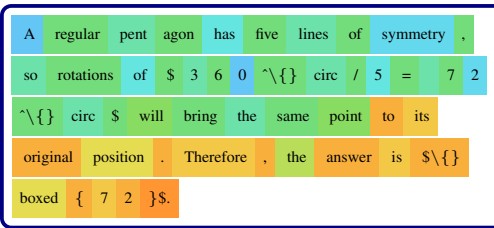

**DUS B=64** (Steps: 7, Generation length: 97)

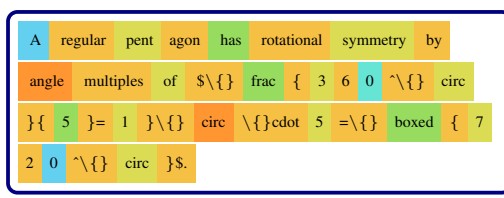

**Fixed B=32** (Steps: 15, Generation length: 72)

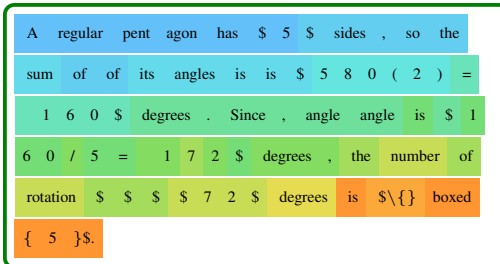

**Fixed B=64** (Steps: 6, Generation length: 25)

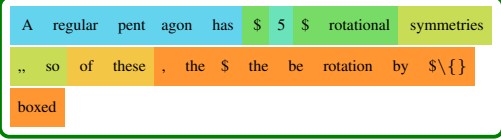

## C.2 GSM8K - LLaDA-8B-Instruct

QUESTION (NO FEW-SHOTS)

PROMPT LENGTH: 1078 TOKENS

> Question: Brandon's iPhone is four times as old as Ben's iPhone. Ben's iPhone is two times older than Suzy's iPhone. If Suzy's iPhone is 1 year old, how old is Brandon's iPhone?
> Answer:

**DUS B=32** (Steps: 15, Generation length: 96)

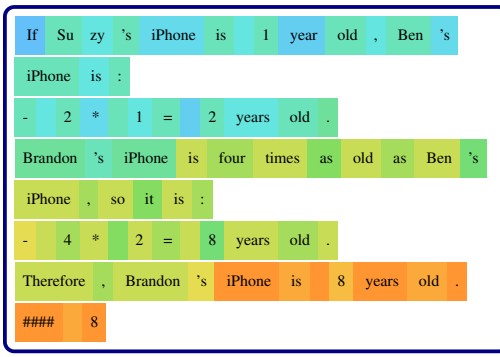

**DUS B=64** (Steps: 12, Generation length: 128)

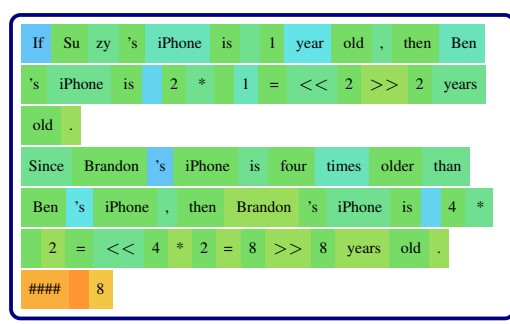

**Fixed B=32** (Steps: 35, Generation length: 198)

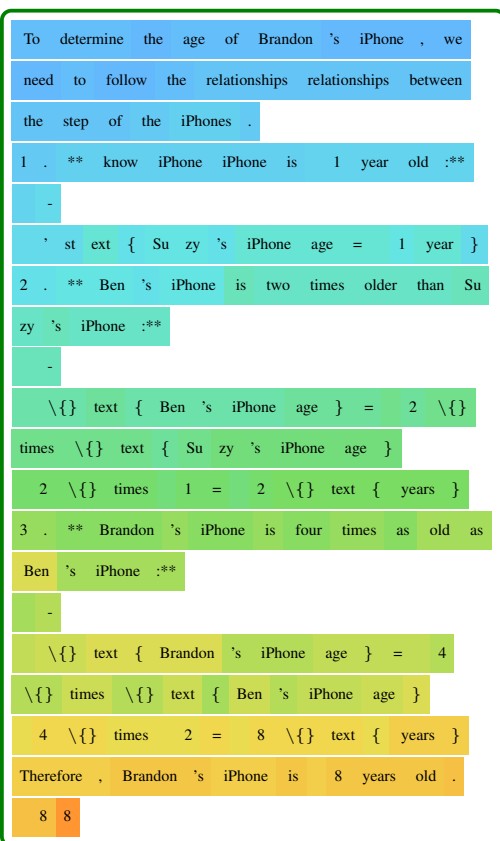

**Fixed B=64** (Steps: 24, Generation length: 253)

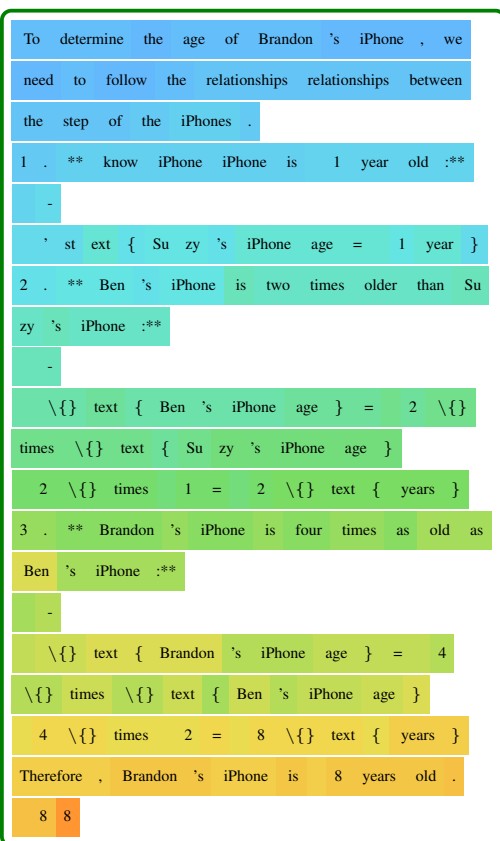

## C.3   HUMANEVAL - LLaDA-8B-INSTRUCT

QUESTION (NO FEW-SHOTS)

PROMPT LENGTH: 71 TOKENS

```
def string_sequence(n: int) -> str:
""" Return a string containing space-delimited numbers starting from 0 upto n inclusive.
>>> string_sequence(0)
'0'
>>> string_sequence(5)
'0 1 2 3 4 5'
"""
```

**DUS B=32** (Steps: 5, Generation length: 32)

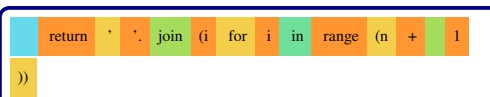

**DUS B=64** (Steps: 7, Generation length: 97)

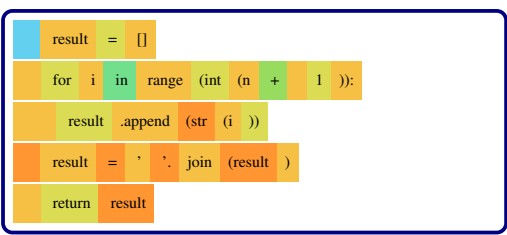

**Fixed B=32** (Steps: 10, Generation length: 44)

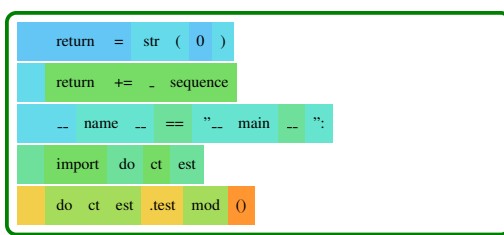

**Fixed B=64** (Steps: 6, Generation length: 19)

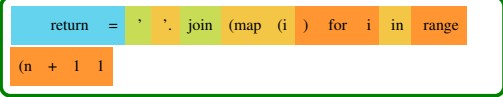

## C.4 MBPP - LLADA-8B-INSTRUCT

QUESTION (NO FEW-SHOTS)

PROMPT LENGTH: 699 TOKENS

> Write a function to find the gcd of the given array elements. Your code should pass these tests:
> ```
> assert get_gcd([2, 4, 6, 8, 16]) == 2
> assert get_gcd([1, 2, 3]) == 1
> assert get_gcd([2, 4, 6, 8]) == 2
> [BEGIN]
> ```

**DUS B=32** (Steps: 10, Generation length: 64)

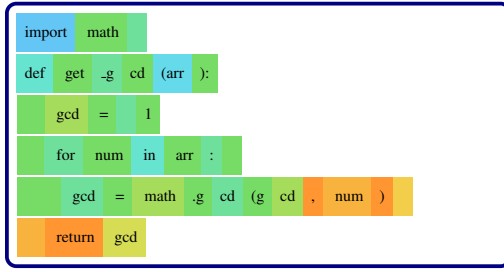

**DUS B=64** (Steps: 7, Generation length: 97)

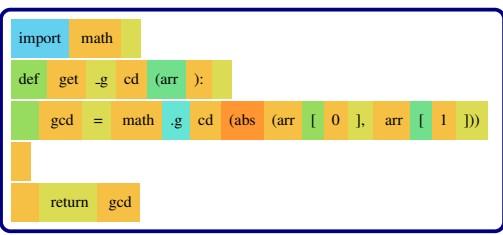

**Fixed B=32** (Steps: 10, Generation length: 53)

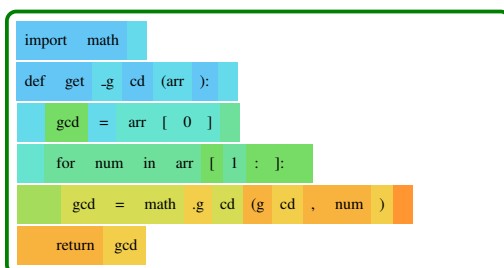

**Fixed B=64** (Steps: 6, Generation length: 30)

```
import math
def get _g cd (arr ):
    gcd = math .g cd (* arr , reduce
)
ONE ]]
```

## C.5 MMLU-PRO BIOLOGY - LLADA-8B-INSTRUCT

QUESTION (NO FEW-SHOTS)

PROMPT LENGTH: 1400 TOKENS

> Question:
> Which of the following statements is NOT correct about apoptosis?
> Options:
> A. Apoptosis, a special type of cell division, requires multiple cell signaling.
> B. The fact that apoptosis is widespread across several kingdoms is evidence that it evolved early in the evolution of eukaryotes.
> C. Apoptosis plays a crucial role in the development of fingers in embryonic development.
> D. Apoptosis prevents an aging and dying cell from damaging neighboring cells.

**DUS B=32** (Steps: 15, Generation length: 96)

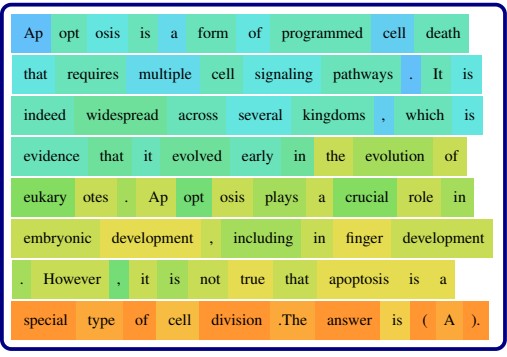

**DUS B=64** (Steps: 13, Generation length: 161)

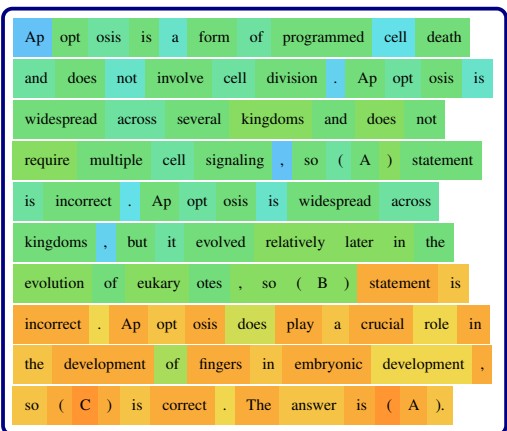

**Fixed B=32** (Steps: 40, Generation length: 256)

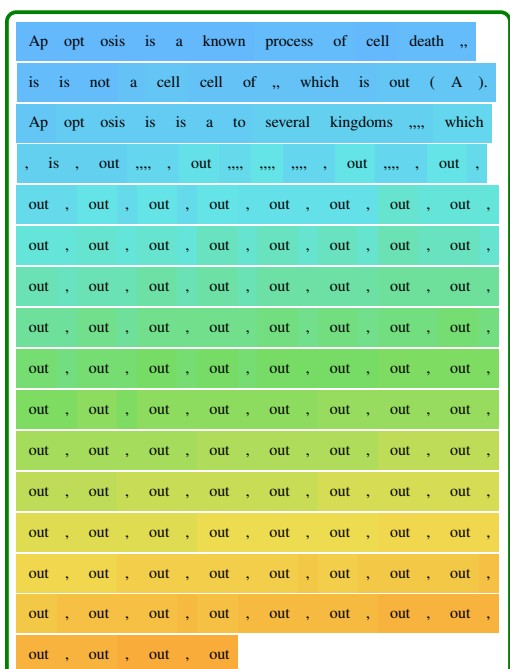

**Fixed B=64** (Steps: 6, Generation length: 44)

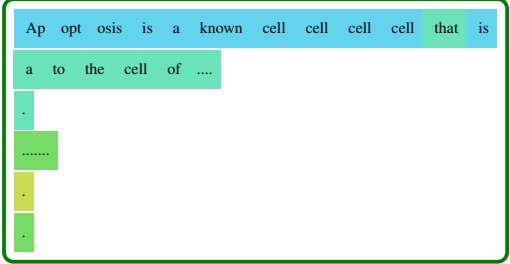

