# OpenReview forum: "Plan for Speed: Dilated Scheduling for Masked Diffusion Language Models"
_ICLR.cc/2026/Conference — Submitted to ICLR 2026_

### Official Review · Reviewer_Shxs · 2025-10-28

**Soundness:** 3
**Presentation:** 1
**Contribution:** 2
**Rating:** 4
**Confidence:** 4

**Summary:**

This paper proposes a fixed schedule for sampling masked diffusion models whereby an exponentially increasing number of tokens are revealed at each step. The tokens are spaced maximally apart to reduce the mutual information between the revealed positions, thus incurring less error when sampling versus a token-by-token sampling approach. The authors demonstrate improved performance per NFE versus choosing token positions to reveal via the common self-confidence metric.

**Strengths:**

The idea is quite simple and easy to implement so could be immediately used by practitioners to improve performance. The experiments show strong results with across the board improvements on practically relevant large scale models on tasks that I believe ICLR attendees will care about e.g. math and coding benchmarks. The authors include comparisons to a self-confidence planner and to a random position planner in the Appendix, both of which are important ablations.

**Weaknesses:**

The paper's main weakness is in the presentation of the motivation and theoretical analysis of the dilated schedule. I find the description hard to follow and it lacks a lot of detail. Specifically, I have the following doubts:

- In equation 2 it is never justified why the minimization objective is the joint entropy of the full sequence. The joint entropy is not even defined at this point. I assume the authors refer to the entropy of the denoising distribution but this is not clear. It is also not clear why we would want to minimize the entropy of the denoising distribution and why this leads to better samples. I think the author's may be referring to the expected log-likelihood assigned by the model to the datapoints (i.e. the expectation is taken with respect to the data distribution and the probabilities are assigned by the model, a type of cross entropy). You would then want a schedule that maximizes the expected log-likelihood.
- L239 is unclear when referring to the self-confidence planner. This method will select the k-top positions with the highest model likelihood, however the authors simply write argmax p_D(X_{i_j} | S_t) for j \in \{1, \dots, k\} which is not clear. The argmax needs to be taken over positions not already selected as part of the top k and the indeces over which the argmax is taken should be specified.
- Equation (5) is not specific to the self-confidence planner, in fact it doesn't use any property of the self-confidence planner, it is true for any planner, including the schedule the authors propose. It doesn't make sense to use equation (5) as a justification for the limitations of the self-confidence planner.
- In L325 with the use of the markov property, there should be justification that this is valid for any indexing i_j of the markov chain. The markov property is usually specified for a certain ordering, so there should be justification that it is also valid for any other ordering of the chain.
- L331 is very inexact, with the whole proof hinging on removing an epsilon. If L331 was true, then the author's method would have the same accuracy as token-by-token unmasking due to the conditional entropy being equal to the sum of factorized entropies. This is clearly not the case, see Figure 1(a). Therefore Equation (12) is not capturing the effect the authors want it to. I understand the motivation in that picking far away positions should reduce their correlation and so reduce the error incurred when revealing them together. However the current 'proof' of this is not at all up to the required standard for a conference paper.
- The authors rely on the fast-mixing properties of markov chains for their proof so this property should be investigated for the datasets that the authors test on. In fact, I would be very surprised if this assumption held true for complex datasets such as code for which distant tokens can very much have a high mutual information (e.g. variable definitions and uses of the variable in far apart places in code).
- I think the ablation in A.2 where the authors compare to a random planner is important and should be included in the main text. The authors hand craft a schedule which reveals less tokens at the start and more tokens at the end for this baseline, but this is already well known as a 'cosine' style schedule that performs the best in prior work [1]. The authors should have just used the cosine schedule as the baseline.

Overall, I think the experimental results are strong but the entirety of section 3 is a poor explanation of the method so much so that I don't think I can recommend this paper for acceptance.

References
[1] Simplified and Generalized Masked Diffusion for Discrete Data, Shi et al. https://arxiv.org/pdf/2406.04329

**Questions:**

If you have a fixed revealing schedule that is always used at inference time, why not train the model using only that schedule as well? There is no point training the model with all of these different possible revealing orderings when at inference time you only ever choose one of them.

Where does the k-1 term come from in equation (11), it doesn't seem to appear anywhere else in Lemma 1.

---

> ### Author Response · Authors · 2025-11-23
>
> We appreciate the reviewer’s thorough analysis and use the points raised to substantially refine the theory section, as detailed below.
>
> ---
>
> 1. **Motivation for Equation (2) and the planner objective**
>
> We agree that the original presentation of Eq. (2) was unclear. In the revised Method section we now explicitly define, for each planner $P$, the planner-induced sampling distribution $q_{\theta,P}(X \mid S_0)$, and the corresponding joint conditional entropy
> $$
> H_{\theta,P}(X \mid S_0) = \mathbb{E}_{q_{\theta,P}}\big[-\log q_{\theta,P}(X \mid S_0)\big].
> $$
> At inference time the denoiser parameters $\theta$ are fixed; the only degree of freedom is the planner.
>
> KL between data and model, or between a slow and fast sampler, is not tractable here because we do not have access to those full distributions. In contrast, $H_{\theta,P}(X \mid S_0)$ is exactly the expected negative log-likelihood under the planner-dependent sampling process, so minimizing this entropy is a natural objective in this setting.
>
> ---
>
> 2. **Clarifying the self-confidence planner (Line 239)**
>
> We agree the original description around Line 239 was imprecise. In the revised text we now define a general score $s(i \mid S_t)$ computed from $p_\theta(X_i \mid S_t)$, and explicitly state that, at each iteration, the self-confidence planner selects the top-$k$ masked indices according to this score (or those above a threshold), always from the set of still-masked positions.
>
> This removes the ambiguity in the earlier informal “$\arg\max$” notation and makes the index set and selection rule precise.
>
> ---
>
> 3. **Equation (5) not specific to the self-confidence planner**
>
> We agree that the basic entropy inequality itself is not specific to self-confidence. In the revision we move this bound into the general “Detailed Formulation” subsection and explicitly state that
> $$
> H_\theta(X_{i_1},\dots,X_{i_k} \mid S_t) \le \sum_{j=1}^k H_\theta(X_{i_j} \mid S_t)
> $$
> holds for any planner that unmasks $k$ tokens in parallel.
>
> We then clarify that the role of DUS is not to change this inequality, but to tighten the gap between the intractable joint entropy (LHS) and the factorized surrogate (RHS) by spacing tokens so that mutual information between simultaneously updated positions is small under the fast-mixing model.
>
> ---
>
> 4. **Use of the Markov property for arbitrary indexings (Line 325)**
>
> We agree that the original proof used the Markov property in a way that was not clearly justified for arbitrary index sets $\{i_j\}$. In the revised version we address this in two ways:
>
> 1. We no longer rely on “Markov for arbitrary indexing.”
>    The previous step that implicitly treated $(X_{i_1},\dots,X_{i_k})$ as a first-order Markov chain and replaced $I(X_{i_j};X_{i_1},\dots,X_{i_{j-1}} \mid S_t)$ by $I(X_{i_j};X_{i_{j-1}} \mid S_t)$ has been removed. We do not assume that the subsequence picked by the planner is itself Markov.
>
> 2. We use only distances + MI decay.
>    In the new proof we fix an ordering $1 \le i_1 < \cdots < i_k \le B$ for convenience, but the argument depends only on the pairwise distances $|i_m - i_n|$. The MI decay lemma for fast-mixing variable-length Markov chains is applied as a function of these distances. As a result, the bound on joint MI holds for any index set whose pairwise distances exceed the threshold.
>
> Thus the revised argument no longer extends the Markov property to an arbitrary reindexing; it uses the variable-length Markov + fast-mixing assumption only through the standard distance-based MI decay.
>
> ---
>
> 5. **Approximate equality and the $\varepsilon$ term (Line 331 / Eq. (12))**
>
> We agree that the original text around Line 331 overstated the result by informally writing an approximate equality and “removing” $\varepsilon$. In the revised theory section we no longer claim equality. Instead, Lemma 1 and its proof now rigorously show that for any $\varepsilon > 0$ there exists a distance threshold $D_\varepsilon$ such that, if all selected indices are at least $D_\varepsilon$ apart, then
> $$
> H_\theta(X_{i_1},\dots,X_{i_k} \mid S_t) \ge \sum_{j=1}^k H_\theta(X_{i_j} \mid S_t) - \varepsilon.
> $$
> We explicitly keep the $\varepsilon$ term and interpret the result as “DUS can make the gap arbitrarily small under the fast-mixing model,” rather than ever equating the joint and factorized entropies. This avoids the incorrect implication that DUS must match token-by-token accuracy.
>
> ---

---

> > ### Author Response · Authors · 2025-11-23
> > **(Continue)**
> >
> > 6. **Fast-mixing assumption vs. real datasets**
> >
> > We agree that the fast-mixing Markov model is an idealized assumption and that real language and code can exhibit strong long-range dependencies (e.g., distant variable uses or rhyme schemes). In the revised Method section we now state explicitly that the stationary, ergodic, fast-mixing chain is used as an analytical model for the sequence along the index dimension, intended to capture that many dependencies are local and that correlations often decay with distance, but not as a claim that real data strictly satisfy the same MI decay everywhere.
> >
> > We also added a dedicated discussion paragraph noting that in regimes with especially strong long-range structure any parallel unmasking scheme, including DUS, can still unmask such tokens in parallel and incur larger entropy gaps. At the same time, because DUS keeps updates spread out and many of our tasks admit multiple valid completions (different correct programs, proofs, or phrasings), later iterations can repair local inconsistencies around early parallel choices by adjusting neighboring tokens. Empirically, on code and math benchmarks that do contain such long-range constraints, DUS still consistently improves over the standard self-planner at matched speedups, which suggests that the qualitative guidance from the fast-mixing model remains useful even when the assumption is only approximate.
> >
> > ---
> >
> > 7. **Random planner ablation, cosine schedules, and placement in the paper**
> >
> > We appreciate this suggestion. In the revised version we highlight the random / non-uniform planner ablation in the main text and clarify its relationship to cosine-style schedules. The baseline used is intentionally designed to reveal fewer tokens early and more later, in line with the cosine-like intuition from prior work, but matching the exact amount of DUS for fair comparison. Even under this stronger, non-uniform baseline, DUS yields higher accuracy at the same nominal speedups, indicating that the gains are not just from “any” non-uniform schedule but from the dilated spacing pattern that the theory motivates. We now make this positioning explicit in the main text.
> >
> > ---
> >
> > 8. **Training-time vs. inference-time schedules**
> >
> > We agree that training-time schedule alignment with a specific planner is an interesting direction. Recent masked diffusion LM papers train with stochastic masking schedules and show that these models are robust to many possible unmasking orders and continue to improve as training proceeds on repeated passes over the same corpus, with loss following smooth scaling laws rather than quickly overfitting. This suggests that, in practice, the denoiser can tolerate a wide range of inference-time planners without requiring schedule-aligned retraining.
> >
> > Our contribution is on the inference side: DUS is meant to demonstrate that purely denoiser-guided self-planners (confidence/entropy based) are not necessarily optimal, and that planner design remains an open problem even for a fixed, well-trained diffusion LM. Diffusion LMs already enjoy advantages such as any-order modeling and strong multi-token prediction, making them a plausible path to mitigate some autoregressive bottlenecks targeted by methods like speculative decoding. DUS fits into this picture as an inference-time scheduling rule that improves the speed–quality tradeoff of existing masked diffusion LMs.
> >
> > ---
> >
> > 9. **Origin of the $k-1$ term in Equation (11)**
> >
> > Lemma 1 has been restated, and we hope it is now clearer. Equation (9) may raise the same question, and the origin of the $k-1$ in the denominator comes from the following transition: for any fixed $k$ and any $\varepsilon > 0$, there exists $d_0(k)$ such that for all $d \ge d_0(k)$,
> > $$
> > C \rho^d \le \frac{\varepsilon}{(k-1)^2},
> > $$
> > which connects the new lemma to the bound on the joint mutual information and explains the appearance of the $(k-1)$ factor.

---

### Official Review · Reviewer_udB1 · 2025-10-30

**Soundness:** 3
**Presentation:** 3
**Contribution:** 2
**Rating:** 2
**Confidence:** 4

**Summary:**

This paper proposes Dilated Unmasking Scheduler (DUS), an inference-only decoding strategy for masked diffusion language models (MDLMs). The core idea is to decode tokens in dilated groups—nonadjacent positions unmasked in parallel—to balance speed and quality. The authors provide a theoretical justification under a Markov assumption that DUS minimizes joint entropy, and empirically evaluate it on reasoning (GSM8K, MATH500), coding (HumanEval, MBPP), and general-knowledge (BBH, MMLU-Pro) benchmarks using diffusion-based LLMs such as LLaDA, Dream, and DiffuCoder.

**Strengths:**

The topic—efficient decoding for diffusion-based LLMs—is timely and relevant, especially as diffusion LMs mature.

The proposed method is inference-only, model-agnostic, and easy to integrate, which has potential practical utility.

The paper is well-organized and clearly written, with formal derivations and visualizations supporting the intuition of DUS.

**Weaknesses:**

- Fundamental limitation unaddressed: token independence in parallel decoding. The paper’s core problem is conceptual. When decoding multiple tokens simultaneously, those positions are inherently independent given the model formulation. Regardless of scheduling or dilation, unmasking multiple tokens at once means the model predicts each without access to the others’ values. DUS does not fundamentally overcome this limitation—it merely spreads out the unmasked tokens in a deterministic pattern.
Thus, the claim that it “minimizes joint entropy” or captures inter-token dependencies is theoretically weak and, in practice, illusory. The independence assumption is the bottleneck of all parallel decoding methods, and this paper does not offer a real mechanism to break it.

- Overstated novelty relative to prior blockwise decoding. The proposed “dilated schedule” is essentially a fixed blockwise unmasking pattern, which has been explored extensively (e.g., LLaDA’s semi-AR decoding, and block diffusion). The difference from prior work is largely heuristic rather than conceptual. Calling it a “new planner” seems overstated; it is better viewed as a deterministic variant of blockwise decoding with pre-specified indices.


Experimental evaluation is superficial

- The experiments are limited to basic benchmarks (GSM8K, MBPP, etc.)
- Crucially, the authors omit a baseline comparison with vanilla token-by-token diffusion sampling. Without showing the raw performance degradation when applying DUS versus standard sequential decoding, it is impossible to quantify the real trade-off between speed and accuracy.
- Missing ablations: results with different diffusion step counts (NFEs), qualitative outputs, or comparisons to autoregressive baselines (e.g., LLaMA-3 or Qwen) are incomplete.

**Questions:**

see weaknesses.

---

> ### Author Response · Authors · 2025-11-23
>
> We thank the reviewer for the thoughtful review and address the primary comments below.
>
> ---
>
> 1. **Fundamental limitation: independence in parallel decoding**
>
> We agree that parallel diffusion decoding, by construction, predicts multiple tokens conditionally independently given $S_t$, and that DUS does not change the denoiser architecture or “break” this limitation. Our contribution is not to alter the model class, but to study how planner design affects the error induced by this parallelism.
>
> In the revised paper we have toned down and clarified the claims: we no longer state that DUS “minimizes the joint entropy” in an absolute sense. Instead, under the fast-mixing Markov model, we show that DUS is designed to (i) keep the gap between the intractable joint conditional entropy $H(X_{i_1},\dots,X_{i_k} \mid S_t)$ and its standard factorized surrogate $\sum_j H(X_{i_j} \mid S_t)$ small at each step by spacing updates, and hence (ii) reduce the cumulative entropy (and error) accumulated by parallel decoding compared to planners that repeatedly pick nearby tokens.
>
> DUS thus does not eliminate conditional independence in the denoiser, but mitigates its negative effects: it spreads updates across the sequence, defers uncertain tokens via the skip mechanism, and empirically yields higher accuracy / Pass@1 than the standard self-planner at the same speedups on GSM8K, HumanEval, and MBPP. We now make this distinction explicit in Section 3 and in the theory discussion.
>
> ---
>
> 2. **Novelty relative to prior blockwise decoding**
>
> We agree that DUS operates within a blockwise, semi-AR decoding framework, and we have clarified this positioning in the revised text. The contribution is not a new blockwise architecture, but an inference-time planner that specifies which positions to reveal under an existing masked diffusion LM. Blockwise diffusion and LLaDA-style semi-AR decoding modify the training objective or architecture to generate spans jointly; DUS leaves the denoiser unchanged and only changes the ordering and spacing of unmasking decisions.
>
> Conceptually, DUS differs from standard blockwise schedules in two ways:
>
> 1. It uses a dilated, log-depth schedule that is explicitly designed (and theoretically analyzed) to keep simultaneously updated tokens far apart, tightening the usual joint-vs-factorized entropy approximation.
>
> 2. It includes a skip mechanism that allows high-uncertainty tokens to be deferred to later iterations without extra NFEs, making it partially adaptive rather than a purely fixed mask pattern.
>
> We have updated the paper to emphasize that DUS is an inference-only planning strategy built on top of existing blockwise diffusion decoders, and we avoid phrasing that suggests a new architectural family.
>
> ---
>
> 3. **Scope of experimental evaluation**
>
> The aim in this work is to study planners for masked diffusion LMs specifically in structured reasoning and code regimes, where semi-AR diffusion decoders are already widely used and speed–quality tradeoffs are especially critical. For that reason the experiments focus on six benchmarks that cover three such domains: two code-generation tasks (HumanEval, MBPP), two math/logic tasks (GSM8K, SVAMP), and two general reasoning-style evaluations.
>
> ---
>
> 4. **Missing sequential diffusion baseline**
>
> We agree that a comparison to vanilla token-by-token diffusion sampling is important to quantify the speed–accuracy tradeoff. In the revised version we now include the standard sequential (token-by-token) diffusion baseline for each model and dataset, reported side by side with the self-planner and DUS results. This makes the performance gap to the full sequential sampler explicit in the main tables and figures.
>
> ---
>
> 5. **Additional ablations and autoregressive baselines**
>
> We thank the reviewer for the suggestions on additional ablations and baselines. The focus here is on comparing planners under a fixed diffusion LM, rather than exploring the full NFE or architecture tradeoff space. In this setting, the effective speedup is directly controlled by the semi-AR block size and schedule (how many tokens are unmasked per iteration and the total generation length), and all reported results are organized by this speedup factor.
>
> For the new EB-style and threshold baselines that we are adding, we also report their effective NFEs alongside the same speedup factors so that cross-sampler efficiency can be compared more transparently.
>
> Regarding autoregressive models, the paper does not claim that DUS makes diffusion LMs surpass strong AR systems. Progress on absolute performance is largely driven by denoiser design and training (e.g., LLaDA, Dream, DiffuCoder), whereas the contribution is orthogonal: improving the speed–quality tradeoff for a given masked diffusion LM via a better planner. The updated experiments show that DUS consistently improves over the standard self-planner at matched speedups.

---

> > ### Comment · Reviewer_udB1 · 2025-11-26
> >
> > thank you for your response. Given that the authors do not conduct more experiments requested I will maintain my score.

---

> > > ### Author Response · Authors · 2025-11-27
> > >
> > > We thank the reviewer for the fast response. We would like to clarify that **the main experimental requests from the original review have now been addressed in the revised version**, and to make explicit how each request maps to the new results.
> > >
> > > 1. **Sequential diffusion (token by token) baseline**
> > > The review noted the absence of a vanilla token by token diffusion baseline. In the revised paper, we now include the standard sequential diffusion sampler for every model and benchmark, reported explicitly in the main tables with its own dedicated row. This makes the accuracy gap to full sequential diffusion clear for GSM8K, MATH500, HumanEval, MBPP, BBH, MMLU-Pro, and IFEval, and allows a direct comparison against the self-planner and DUS.
> > >
> > > | Model          | GSM8K | MATH500 | HumanEval | MBPP  | BBH   | MMLU-Pro |
> > > |---------------|:-----:|:-------:|:---------:|:-----:|:-----:|:--------:|
> > > | LLaDA-B       | 72.63 |  24.00  |   34.76   | 38.00 | 44.26 |  39.82   |
> > > | LLaDA-I       | 80.29 |  28.80  |   39.02   | 39.40 | 53.89 |  40.89   |
> > > | Dream-I       | 77.10 |  37.00  |   57.90   | 56.20 | 58.15 |  50.89   |
> > > | DiffuCoder-B  |     |       |   67.10   | 74.20 |     |         |
> > > | DiffuCoder-I  |      |        |   72.00   | 75.10 |     |         |
> > >
> > > (Entries are the ×1 token-by-token diffusion scores.)
> > >
> > > 2. **Stronger planner baselines and NFE / speed–quality ablations (EB-Sampler, CB-Sampler, parallelism sweeps)**
> > > The review requested stronger and more modern planning baselines, as well as more systematic ablations over NFEs and speed–quality tradeoffs. In response, we **(a)** implemented EB-Sampler and CB-Sampler according to [1,2], **(b)** evaluated them on IFEval benchmark together with DUS, and **(c)** reported their task scores against the effective number of denoising function evaluations (NFEs), including sweeps over the confidence and entropy thresholds, yielding explicit speed–quality curves. Together, these results directly address the request for more ablations on NFEs and parallel decoding tradeoffs.
> > >
> > > We refer to Section 4.3 and specifically to Figure 3 for more details
> > >
> > > 3. **AR baselines**
> > > The review pointed out the lack of autoregressive baselines such as LLaMA or Qwen. In the revised version, we now include AR baselines with their own dedicated rows in each relevant table. This makes the absolute performance gap between current diffusion LMs (with different planners) and strong AR models visible, even though our primary focus is the relative behavior of planners under a fixed diffusion LM.
> > >
> > > | Dataset (G)      | Llama3-8B | Qwen3-8B |
> > > |------------------|:---------:|:--------:|
> > > | GSM8K (G=256)    |   49.81   |  88.55   |
> > > | MATH500 (G=256)  |   15.23   |  50.20   |
> > > | HumanEval (G=512)|   36.59   |  61.59   |
> > > | MBPP (G=512)     |   48.40   |  65.40   |
> > > | MMLU-Pro (G=256) |   37.86   |  52.68   |
> > > | BBH (G=256)      |   52.59   |  59.63   |
> > > | IFEval (G=1024)  |   15.23   |    41.73     |
> > >
> > > 4. **Qualitative case studies**
> > > Finally, the review requested more qualitative outputs. Due to space limits these are not in the main text, but we now provide qualitative case studies comparing planners in Appendix C, which illustrate the kinds of errors introduced or avoided by different schedules.
> > >
> > > We hope this makes it clear that the revised submission incorporates the key experimental elements requested in the original review: **sequential diffusion baselines** across all benchmarks, **EB-Sampler and CB-Sampler baselines [1,2] with NFE-aware speed-quality curves on IFEval**, **AR baselines**, and **qualitative outputs in the appendix**.
> > >
> > >
> > > [1] Wu et al. Fast-dLLM: Training-free Acceleration of Diffusion LLM by Enabling KV Cache and Parallel Decoding. arXiv 2025.05.
> > >
> > > [2] Ben-Hamu et al. Accelerated Sampling from Masked Diffusion Models via Entropy Bounded Unmasking. arXiv 2025.05.

---

### Official Review · Reviewer_R3sn · 2025-11-02

**Soundness:** 2
**Presentation:** 1
**Contribution:** 2
**Rating:** 2
**Confidence:** 4

**Summary:**

This paper proposes the Dilated Unmasking Scheduler (DUS), an inference-only method for accelerating masked diffusion language models. DUS divides sequence positions into non-adjacent dilated groups and unmasks them in parallel to minimize an upper bound on joint entropy gain at each step. Experiments on math (GSM8K, MATH500), code (HumanEval, MBPP), and reasoning benchmarks (BBH, MMLU-Pro) show that DUS outperforms confidence-based planners under the same compute budget.

**Strengths:**

1. The method is entirely training-free, requiring no model modification or additional supervision, which makes it simple and practical for accelerating MDLM inference.

2. The authors conduct experiments on a broad and representative set of benchmarks covering mathematical reasoning, code generation, and general knowledge tasks, demonstrating the method’s general applicability.

**Weaknesses:**

1. **Poor citation formatting.**
The paper consistently uses `\citet{}` where `\citep{}` should be applied.

2. **Confused mathematical reasoning.**
   (1) In Lines 160–161, the authors assume that $X$ is a sequence of random variables forming a stationary, ergodic, first-order Markov chain with fast-mixing properties, but provide no explanation of the validity or relevance of this assumption.
   (2) The paper adopts **minimizing conditional entropy** as the optimization objective but does not justify why this objective is reasonable. Typically, minimizing the KL divergence between data and model distributions, or between the distributions induced by the original slow sampler and the new fast sampler, is more common. The rationale for using conditional entropy minimization needs further clarification.
   (3) The authors derive that the conditional entropy of the confidence-based sampling method satisfies $\le \sum_{j=1}^k H(X_{i_j}|S_t)$, while that of the proposed DUS method approximately satisfies the same inequality. Even if minimizing conditional entropy is the goal, it is unclear why this theoretical result implies that the proposed method performs better.
(4) In addition, the definition of $H(X|S_0)$ in Eq. (2) is never introduced. Since the distribution $p_{\theta}(X|S_0)$ is fixed once the model is trained, it is unclear why $H(X|S_0)$ can be treated as an optimizable quantity.

3. **Outdated baseline.**
   Widely used confidence- or entropy-threshold-based methods can already achieve nearly lossless 2–3× acceleration [1, 2]. However, the authors choose only the earliest confidence-based acceleration strategy, which suffers severe performance degradation even at a 2.7× speedup.

[1] Wu et al. Fast-dLLM: Training-free Acceleration of Diffusion LLM by Enabling KV Cache and Parallel Decoding. arXiv 2025.05.

[2] Ben-Hamu et al. Accelerated Sampling from Masked Diffusion Models via Entropy Bounded Unmasking. arXiv 2025.05.

**Questions:**

Please refer to weaknesses.

---

> ### Author Response · Authors · 2025-11-23
>
> We appreciate the reviewer’s detailed evaluation and respond to the main concerns point by point.
>
> ---
>
> 1. **Citation formatting**
>
> We thank the reviewer for pointing this out. In the revised version we systematically correct the citation formatting.
>
> ---
>
> 2. **Confused mathematical reasoning**
>
> **(1) Markov + fast-mixing assumption**
>
> We agree this assumption needed to be motivated. In the revised Method section (Notation + Theory) we now explicitly state that the stationary, ergodic, fast-mixing Markov chain is a stylized model used only for analysis, meant to capture that in many text domains local dependencies dominate and correlations typically decay with distance. We also explicitly discuss edge cases (rhyme, code, long-range constraints) where this model is violated and treat the theory as qualitative guidance for why spacing out updates can help.
>
> **(2) Why conditional entropy, not KL**
>
> At inference time the denoiser parameters $\theta$ are fixed; the only degree of freedom is the planner $P$. For a given planner, running the sampler induces a distribution $q_{\theta,P}(X \mid S_0)$ over completions. We now define
> $$
> H_{\theta,\mathcal{P}}(\mathcal{X} \mid S_0) = \mathbb{E}{q{\theta,\mathcal{P}}}[-\log q_{\theta,\mathcal{P}}(\mathcal{X} \mid S_0)],
> $$
> i.e., the expected negative log-likelihood under the planner-dependent sampling distribution. KL between data and model, or between slow and fast samplers, is not tractable at inference because we do not have access to the data distribution or to the full slow-sampler distribution. In contrast, conditional entropy is directly tied to the log-likelihood under $q_{\theta,P}$ and varies with $P$, so minimizing $H_{\theta,P}(X \mid S_0)$ is a well-defined planner objective under a fixed denoiser.
>
> **(3) Why DUS vs confidence-based planner**
>
> The inequality itself holds for any planner that unmasks $k$ tokens in parallel; we now clarify that the difference comes from how tight the usual surrogate $\sum_j H_\theta(X_{i_j} \mid S_t)$ is to the true joint entropy $H_\theta(X_{i_1},\dots,X_{i_k} \mid S_t)$. Standard self-planners choose tokens solely by per-token confidence/entropy and can repeatedly select nearby positions, where mutual information is large and the surrogate can severely overestimate the joint entropy. Under the Markov + fast-mixing model, DUS keeps simultaneously updated positions well separated, which (by Lemma 1 + the MI decay lemma) makes the gap between the joint entropy and its surrogate arbitrarily small. We have rewritten this part of the theory section to emphasize that DUS is designed to minimize this gap, not just satisfy the same inequality.
>
> ---
>
> 3. **Outdated baseline and threshold-based accelerations**
>
> We thank the reviewer for pointing out these recent threshold-based accelerations. Fast-dLLM and the EB-sampler both adaptively choose how many tokens to unmask per step based on confidence/entropy thresholds, effectively varying the total number of denoiser calls. In contrast, DUS is a planner that provides a fixed, training-free $\mathcal{O}(\log B)$ schedule for a given semi-AR block size, with a deterministic NFE and speedup. These approaches therefore optimize slightly different objectives: adaptive methods trade off quality against a variable NFE budget, whereas DUS targets a guaranteed speedup at fixed NFE and focuses on how to best allocate the parallel updates.
>
> We have implemented EB-style and confidence-threshold baselines following Wu et al. and Ben-Hamu et al. and are running them in our setup; their results will be included alongside DUS in the updated experiments. In addition, the concurrent work *Parallel Sampling from Masked Diffusion Models via Conditional Independence Testing* already provides a direct empirical comparison between EB-style samplers and DUS-type dilated planners, further situating our method within this line of threshold-based accelerations.

---

### Official Review · Reviewer_wNci · 2025-11-02

**Soundness:** 3
**Presentation:** 3
**Contribution:** 3
**Rating:** 6
**Confidence:** 3

**Summary:**

The paper introduces Dilated Unmasking Scheduler (DUS) to speed up masked diffusion LMs by revealing non-adjacent tokens in parallel using a coarse-to-fine, dilated pattern (groups roughly double each round). Under a first-order Markov assumption, this fixed schedule aims to minimize joint entropy per step; it’s inference-only (no retraining) and contrasts with learned planners like LLaDA or DDPD. Empirically, DUS completes decoding in O(log N) steps, recovers most quality lost by naive parallelism, and beats confidence-based planners on GSM8K, HumanEval, and MBPP at matched speedups—approaching sequential/AR quality while much faster. Net: a model-agnostic, principled scheduling rule that narrows the parallel–sequential gap.

**Strengths:**

The proposed dilated unmasking schedule is grounded in an information-theoretic rationale rather than ad-hoc heuristics. By maximizing the distance between concurrently unmasked tokens, DUS minimizes their mutual information and achieves the joint entropy reduction bound at each step under a first-order Markov model This is a more rigorous approach than greedy confidence-based strategies that ignore token interdependencies and it's model-free. DUS is a drop-in decoding method requiring no changes to model architecture or training It can be applied to any masked diffusion LM (e.g. the authors test it on different 7B–8B models) without fine-tuning.

DUS yields logarithmic-time decoding in the sequence length, dramatically cutting the number of denoiser without much degradation of perofrmance. with the same average parallelism, it consistently scores higher than the standard self-confidence unmasking heuristic and nearly recovers the single-token diffusion performance (denoted as SOTA at 1× speed) even when unmasking multiple tokens per step

**Weaknesses:**

The “dilated” schedule, while motivated by theory, ultimately rests on a strong independence assumption that may not hold universally. The analysis assumes a fast-mixing first-order Markov chain for token dependencies. For the selected benchmarks, there's evidence that there no such complex dependencies. Real language sequences can have long-range dependencies (e.g. rhyme schemes in poetry or variable usage in code), perhaps evaluating the model on natural language tasks would give different results. Seems necessary to expand the scope of datasets, and report metrics like perplexity or diversity on large-scale text corpora

The lack of learned planner is less expressive than alternative, also efficient approaches with abilty to learn task-specific planners. Similarly, there's no mechanism for changing a token that has been unmasked. recent approaches to discrete diffusion can address this issue, and can fix a mistaken token in later iterations

**Questions:**

N/A

---

> ### Author Response · Authors · 2025-11-23
>
> We thank the reviewer for the careful and constructive assessment of our submission. Below we address the main comments.
>
> ---
>
> 1. **Fast-mixing / independence assumption and dataset scope**
>
> We agree that real language and code can have significant long-range dependencies, and we do not assume global independence. Our analysis uses a fast-mixing Markov model only to reason about the small set of tokens that are unmasked in the same iteration: the intractable joint conditional entropy $H(X_{i_1}, \dots, X_{i_k} \mid S_t)$ is approximated by the standard surrogate $\sum_j H(X_{i_j} \mid S_t)$. DUS is explicitly designed to make this local approximation more accurate by spacing the selected indices so that, under the Markov + fast-mixing model, the mutual information between simultaneously updated tokens is small and the gap between the joint entropy and its factorized surrogate is minimized. We now state this explicitly in the revised Method section (Sec. 3), together with a discussion of the limitations of those assumptions.
>
> Importantly, the experiments are not restricted to settings with only short-range structure. The evaluation covers GSM8K (math/logic) and HumanEval and MBPP (code), which exhibit exactly the kind of long-range dependencies you mention (e.g., variable usage, control flow, bracket matching). Across all these benchmarks, DUS consistently outperforms the standard self-planner at fixed speedups. This suggests that, even when the fast-mixing assumption is only approximate, the dilated schedule remains beneficial in practice. As explained in Section 3, because DUS keeps updates spread out and many tasks admit multiple valid completions (e.g., different correct programs, proofs, or phrasings), later iterations can adjust neighboring tokens and repair local inconsistencies around early parallel decisions. Extending the evaluation to large-scale natural language corpora with perplexity and diversity metrics is a valuable direction but orthogonal to the present focus on code, math, and reasoning tasks, and is left for future work.
>
> ---
>
> 2. **Fixed (non-learned) planner and lack of explicit revision**
>
> DUS targets a different tradeoff than learned, revising planners: its goal is to provide deterministic speedups with a fixed, training-free schedule. Learned or adaptive remasking planners that revisit tokens can be more expressive, but they typically do so by adding extra denoiser steps, and in practice may require more NFEs than even token-by-token sampling at a given quality level.
>
> In contrast, DUS fixes the number of denoiser evaluations per block to $\mathcal{O}(\log B)$ and guarantees a strict speedup over the standard self-planner, while still exposing uncertain tokens to multiple iterations via the skip mechanism: high-entropy or low-confidence positions can be deferred to later DUS iterations without increasing the total number of denoiser calls. Explicit remasking / edit mechanisms from recent work are orthogonal to our contribution and could be combined with DUS in future work if additional NFEs are allowed.

---

### Author Response · Authors · 2025-11-25
**Updated IFEval benchmark**

We have added a new experiment on the IFEval instruction-following benchmark in **Section 4.3** that directly compares DUS with two adaptive parallel schedulers, the entropy-bound sampler (EB-Sampler) and the confidence-bound sampler (CB-Sampler), implemented following the threshold-based strategies of Wu et al. [1] and Ben-Hamu et al. [2] In this experiment, we sweep the respective thresholds and block sizes and report the full speed-quality tradeoff by plotting task score against the total number of denoising function evaluations required to generate a sequence. This new benchmark directly addresses the reviewers’ requests for stronger baselines and explicit speed-quality comparisons (**Reviewer R3sn, W.3; Reviewer udB1, W.5**).

[1] Wu et al. Fast-dLLM: Training-free Acceleration of Diffusion LLM by Enabling KV Cache and Parallel Decoding. arXiv 2025.05.

[2] Ben-Hamu et al. Accelerated Sampling from Masked Diffusion Models via Entropy Bounded Unmasking. arXiv 2025.05.

---

### Author Response · Authors · 2025-11-29
**Author Response Summary (1/2)**

We thank all reviewers for the detailed feedback and constructive suggestions. Across the reviews there is broad agreement on several strengths:

- **Information-theoretic planner:** DUS is grounded in an information-theoretic analysis of parallel decoding.
- **Inference-only, training-free, and model-agnostic:** It is a drop-in decoding method that requires no changes to the denoiser architecture and no additional training or supervision.
- **Strong speedups with minimal performance loss:** The dilated schedule enables logarithmic-time decoding in sequence length and substantially reduces the number of denoiser calls while minimizing the loos in single-token diffusion performance.
- **Broad and practically relevant evaluation:** The method is evaluated on representative benchmarks in math reasoning, code generation, general knowledge reasoning, and instruction following tasks.
- **Simple and easy to adopt:** The method is conceptually simple and easy to implement, so it could be adopted quickly in practice.

---

## Summary of Key Changes

We summarize below the key changes in our rebuttal revision,

- **Theory:** We replaced the first-order Markov assumption with a **Variable-Length Markov Chain (VLMC)** formulation and provided a new proof in Appendix B. We no longer claim an approximate equality between joint and factorized entropy; instead, we now frame DUS as **minimizing the gap** to the ideal joint entropy and establish a rigorous **entropy–likelihood equivalence** for the planner objective.
- **Baselines:** We added **token-by-token diffusion** baselines and strong **autoregressive baselines (Llama-3-8B, Qwen-3-8B)** across all benchmarks (Tables 1–2, Figure 3).
- **Additional benchmarks and planners:** We added the **IFEval** instruction-following benchmark and directly compared DUS to recent adaptive planners **EB-Sampler** and **CB-Sampler**, reporting full **speed–quality tradeoff curves** (Figure 3).
- **Ablations and qualitative analysis:** We moved the **parallelism planner ablations** into the **main text (Section 4.3)** and added **qualitative trajectory visualizations** to Appendix C.

---

## Theoretical Concerns

Reviewers raised questions about the settings of our theoretical claims and its proof. We have accordingly overhauled Section 3,

- **DUS minimizes the gap between joint entropy and the factorized upper bound.**

    The revised theoretical section claims that DUS **minimizes the discrepancy** between the true (intractable) joint entropy and a tractable factorized entropy upper bound. This directly addresses Reviewer Shxs’s critique regarding the validity of the "approximate equality" in the original bound and Reviewer R3sn’s query on why this theoretical result implies better performance.

- **Assumption correction from first-order Markov to VLMC.**

    The strict first-order assumption has been relaxed to a **Variable-Length Markov Chain (VLMC)** one. Appendix B now contains a complete proof showing that rapid decay of mutual information, required for our spacing strategy, holds for VLMCs under fast-mixing conditions. This change directly resolves Reviewer wNci’s worry about overly strong independence assumptions and Reviewer R3sn’s and Reviewer Shxs's criticism of the original first-order assumption.

- **Clarified motivation for minimization term.**

    A new formal proposition establishes that minimizing joint entropy under the planner-induced distribution is **equivalent** to maximizing the expected log-likelihood. This clarifies our objective and answers the questions posed by Reviewer R3sn and Reviewer Shxs regarding the choice of entropy over KL divergence or likelihood.


---

---

> ### Author Response · Authors · 2025-11-29
> **Author Response Summary (2/2)**
>
> ## Experimental Thoroughness
>
> We have substantially broadened and reorganized the experimental section in response to the reviewers’ requests:
>
> - **New benchmark and comparisons to recent adaptive planners.**
>
>     We added the **IFEval** instruction-following benchmark (Section 4.3) and implemented two very recent adaptive planners, **EB-Sampler** and **CB-Sampler** [1,2]. For IFEval we now report full **score vs NFE** curves, giving a detailed view of the speed–accuracy tradeoff. This directly addresses Reviewer R3sn’s concern about “outdated baselines” and the need for comparisons against threshold-based strategies.
>
> - **Autoregressive baselines across all benchmarks.**
>
>     We included size-comparable AR models **Llama-3-8B** and **Qwen-3-8B** and report their performance on every benchmark (Tables 1–2, Figure 3). This provides a clear picture of how DUS-equipped diffusion LMs compare to strong AR systems on math, code, reasoning, and instruction-following tasks, directly addressing Reviewer udB1’s request.
>
>     | Dataset (G) | Llama3-8B | Qwen3-8B |
>     | --- | --- | --- |
>     | GSM8K (G=256) | 49.81 | 88.55 |
>     | MATH500 (G=256) | 15.23 | 50.20 |
>     | HumanEval (G=512) | 36.59 | 61.59 |
>     | MBPP (G=512) | 48.40 | 65.40 |
>     | MMLU-Pro (G=256) | 37.86 | 52.68 |
>     | BBH (G=256) | 52.59 | 59.63 |
>     | IFEval (G=1024) | 15.23 | 41.73 |
> - **Diffusion model baselines: token-by-token diffusion everywhere.**
>
>     A standard **token-by-token diffusion inference** baseline is now reported for all masked diffusion models and datasets (Tables 1–2, Figure 3). This makes the tradeoff between full sequential diffusion and semi-parallel planners fully explicit, as requested by Reviewer udB1.
>
>     | Model (token-by-token) | GSM8K | MATH500 | HumanEval | MBPP | BBH | MMLU-Pro | IFEval |
>     | --- | --- | --- | --- | --- | --- | --- | --- |
>     | LLaDA-B | 72.63 | 24.00 | 34.76 | 38.00 | 44.26 | 39.82 | 35.73 |
>     | LLaDA-I | 80.29 | 28.80 | 39.02 | 39.40 | 53.89 | 40.89 | 62.83 |
>     | Dream-I | 77.10 | 37.00 | 57.90 | 56.20 | 58.15 | 50.89 | 36.81 |
>     | DiffuCoder-B | - | - | 67.10 | 74.20 | - | - | - |
>     | DiffuCoder-I | - | - | 72.00 | 75.10 | - | - | - |
> - **Ablations moved to the main text.**
>
>     Following Reviewer Shxs’s recommendation, the ablation studies comparing **Random vs Self-Confidence** scheduling and **Fixed-k vs Incremental dilation** have been moved from the appendix into **Section 4.3**. These experiments confirm that the gains primarily stem from the dilated structure of the planner, rather than from incidental hyperparameter choices.
>
> - **Qualitative trajectory visualizations.**
>
>     Appendix C now includes detailed **qualitative visualizations of generation trajectories**, illustrating how different planners (self-planner, DUS, EB-Sampler, CB-Sampler) behave on representative math, code, and instruction-following prompts. This addresses Reviewer udB1’s request for more qualitative insight into the types of errors that different schedules induce or avoid.
>
>
> ---
>
> ## Concluding remarks
>
> We are deeply grateful for the reviewers’ thoughtful engagement and constructive feedback. The detailed questions and comments significantly helped us refine the conceptual framing, strengthen the empirical analysis, and improve the clarity of the paper.
>
> Taken together, the move to a VLMC-based analysis, the explicit gap-minimization framing, and the new entropy–likelihood equivalence place the theoretical contribution on much firmer ground and align it with the concerns raised in the reviews. On the empirical side, the expanded evaluation now systematically covers token-by-token diffusion, strong autoregressive baselines, a new instruction-following benchmark (IFEval), and recent adaptive planners (EB-Sampler, CB-Sampler), along with parallelism ablations and qualitative trajectory visualizations.
>
> We hope these revisions address the main theoretical and experimental concerns raised by the reviewers, clarify the scope of our claims, and make the role of DUS as a simple, principled, and practically useful planner for masked diffusion LMs more evident.
>
>
>
> [1] Wu et al. Fast-dLLM: Training-free Acceleration of Diffusion LLM by Enabling KV Cache and Parallel Decoding. arXiv 2025.05.
>
> [2] Ben-Hamu et al. Accelerated Sampling from Masked Diffusion Models via Entropy Bounded Unmasking. arXiv 2025.05.

---

### Meta-Review · Area_Chair_umst · 2025-12-19

**Summary:**

The paper concerns a dilated schedule to decode a block of size B in O(log B) iterations. The main idea is to first decode (in parallel) far away tokens, and hope they are mostly independent, and then to gradually shrink the distance between the tokens that are decoded in log(B) rounds.

While the method is natural and the empirical results seem promising, I agree with the sentiments of multiple reviewers that the paper is not ready for publication for a number of reasons. The mathematical formulation / reasoning seems very confused, as reviewers R3sn and Shxs note. Firstly, it's not really clear *what* conditional entropy the authors are trying to minimize and why --- and the edits that were added after the discussion period do not make sense either. (e.g. equation (2) is clearly wrong as written as it depends only on q but not on the trained decoder --- in which case, just outputting a delta mass with any configuration and completely ignoring the trained decoder will have 0 entropy).  Second, the theory seems detached from the algorithm --- while I agree that if there's sufficiently fast mixing the distribution of the far away tokens will be closer to a product distribution --- at some point, when the fine-grained tokens are getting filled in, they will clearly be very dependent. Thirdly, the experiments are quite preliminary, on a relatively small # of relatively simple benchmarks, and comparing with relatively weak baselines.

**Reviewer Concerns:**

Confused mathematical writing, strong/unreasonable assumptions for the theory, weak experimental setup (relatively simple benchmarks and weak baselines).

**Reviewer Scores:**

I do not think the scores would have updated for the reasons stated in my meta-review.

---

### Decision · Program_Chairs · 2026-01-26

Reject